# Optimized Graph Structures for Calibrating Graph Neural Networks with Out-of-Distribution Nodes

## Abstract

Graph neural networks (GNNs) achieve remarkable success in tasks such as node classification, link prediction, and graph classification. However, despite their effectiveness, the reliability of the GNN's prediction remains a major concern, particularly when graphs contain out-of-distribution (OOD) nodes. To date, the calibration of GNNs in the presence of OOD nodes remains largely under-explored. Our empirical studies reveal that the calibration problem becomes significantly more complex in the presence of OOD nodes, and existing calibration methods are notably less effective in such scenarios. Recently, graph structure learning (GSL), a family of data-centric learning approaches, has shown promise in mitigating the adverse effects of the noisy information in the graph topology by jointly optimizing the graph structure and GNN training. However, current GSL methods do not explicitly address the calibration challenges posed by OOD nodes. To tackle the this challenge, we propose a novel framework called Graph Calibration via Structure Optimization (GCSO) to calibrate GNNs in the presence of OOD nodes. Our empirical findings suggest that reducing the weights of edges connecting in-distribution (ID) and OOD nodes can effectively alleviate the calibration issue. However, identifying such edges and determining their appropriate weights is challenging due to the unknown distribution of OOD nodes. To address this, GCSO introduces an iterative edge-sampling mechanism that captures the topological information of the graph and formulates the structure learning process as a Markov Decision Process (MDP). We then leverage the actor-critic method to dynamically adjust edge weights and evaluate their impact on target node predictions. Additionally, we design a tailored reward signal to guide the policy function toward an optimal graph structure that minimizes the influence of OOD nodes. Notably, our optimized graph structure can be seamlessly integrated with existing temperature scaling-based calibration techniques for further performance gains. Experimental results on benchmark datasets demonstrate that our method significantly reduces the expected calibration error (ECE) while maintaining competitive accuracy. The anonymous GitHub repository for the code is available at `https://anonymous.4open.science/r/calibration-7F61`.

## 1 Introduction

Graph neural networks (GNNs) have proven to be effective for processing graph-structured data, which is prevalent in real-world applications such as social networks, traffic systems, and financial networks. Despite their remarkable success in tasks like node classification, link prediction, and graph classification, the reliability of GNNs has become an increasing concern within the machine learning community. A foundational study (Guo et al., 2017) introduced the expected calibration error (ECE) to quantify the discrepancy between a model's predictive confidence and the actual likelihood of correctness. More recent works (Wang et al., 2021; Hsu et al., 2022; Teixeira et al., 2019; Fang et al., 2024; Yang et al., 2024b) have observed that GNNs often suffer from under-confidence in their predictions.

Existing GNN calibration methods (Wang et al., 2021; Hsu et al., 2022; Fang et al., 2024; Tang et al., 2024) primarily focus on *clean* graphs, where all nodes are assumed to come from the same distribution. However, real-world graphs often contain out-of-distribution (OOD) nodes (Zhao

et al., 2020; Yang et al., 2022; Song & Wang, 2022). For instance, users on social networks may be connected not only to family and friends but also to strangers or even online scammers.

When a graph includes OOD nodes, the calibration of GNNs tends to deteriorate. As illustrated in Fig 1, we designate nodes from specific classes (e.g., last two classes in Cora (Yang et al., 2016)) as OOD nodes, while treating the remaining nodes as in-distribution (ID) nodes. We then perform node classification using GCN (Kipf & Welling, 2016) across several graph benchmark datasets and compare the ECE in scenarios with and without OOD nodes. The results indicate that presence of OOD nodes leads to an increase in ECE. Moreover, the calibration issue becomes more nuanced in the OOD setting. As shown in Fig 2, unlike the general under-confidence observed in GNNs on clean graphs (Wang et al., 2021; Hsu et al., 2022), GNNs in OOD scenarios tend to be over-confident on some nodes and under-confident on others. Our experiments further demonstrate that existing graph calibration methods become less effective when OOD nodes are present.

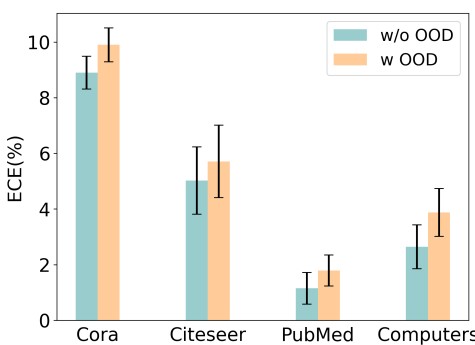

Figure 1: The expected calibration error (ECE) of GCN on graphs with and without OOD nodes.

Recently, graph structuring learning (GSL) (Wu et al., 2022; Zou et al., 2023) has exhibited promising results in mitigating the adverse effects of potential flaws, such as redundant, incorrect or missing connections, by optimizing the graph structure. Recently, Yang et al. (Yang et al., 2024a) proposed the Data-centric Graph Calibration (DCGC) framework which reduces calibration error by modifying the graph structure and assigning higher weights to decisive and homophilic edges. However, this approach does not explicitly consider the out-of-distribution (OOD) scenario. To address calibration in the presence of OOD nodes, Shi et al. (Shi et al., 2023) employed deep Q-learning (Mnih et al., 2013) to calibrate graphs containing OOD nodes. While their method leverages reinforcement learning to modify the graph, it assigns fixed weights to the selected edges, without accounting for variations in the topological structure. Ideally, edge weights should dynamically adapt based on the distribution of OOD nodes. This limitation may lead to suboptimal calibration performance.

Inspired by the prior works (Yang et al., 2024a; Shi et al., 2023) that improve GNN calibration by reweighting edges, we conduct an empirical study and find that the calibration issue can be alleviated to some extent by adjusting the weights of edges connected to OOD nodes. However, identifying the OOD nodes within a graph is not a trivial problem. To address this, we propose Graph Calibration via Structure Optimization (GCSO) which calibrates GNNs through optimized graph structures. Our approach follows the Actor-Critic paradigm. Specifically, We designate labeled nodes as target nodes and sample their adjacent edges. To guide the selection of informative edges, we introduce a novel iterative strategy based on a predefined discrepancy score, formulating the process as a Markov Decision Process (MDP). Next, we leverage the Actor-Critic paradigm to dynamically assess the impact of adjusted edge weights on the target nodes. In particular, we adopt the Deep Deterministic Policy Gradient (DDPG) (Lillicrap et al., 2016) to generate the fine-grained, topology-aware edge weights. In addition, we design a novel reward signal to guide the optimization of edge weights. The reward signal consists of two components: an indicator function and the entropy regularization term for the target nodes. The indicator function aims to preserve the accuracy of the model on node classification with the optimized graph structure, while the entropy regularizes the logit distribution of the target nodes for the calibration purpose. Notably, our optimized graph structure can be seamlessly integrated with the existing post-hoc calibration method to further improve the calibration performance in the downstream tasks. The contribution of this paper is summarized as follows:

- We propose Graph Calibration via Structure Optimization (GCSO) to enhance the calibration of graph neural networks in the presence of OOD nodes. Our approach introduces a novel edge iteration method which is formulated as the Markov Decision Process (MDP). Additionally, A new reward signal is designed to guide the policy function to generate the optimized graph structure that enhances calibration performance.

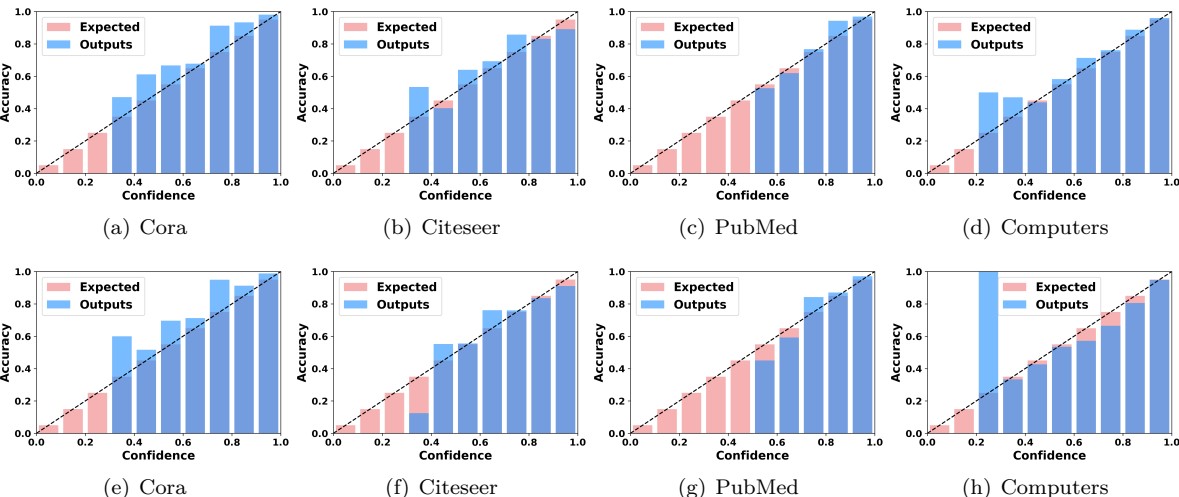

Figure 2: Reliability diagrams of GCN on without OOD nodes ((a)-(d)) and with OOD nodes ((e)-(h)). The results suggest that the calibration issue (i.e., over-confidence or under-confidence) is more complicated with the presence of OOD nodes. Ideally, in a well-calibrated network, accuracy should align with confidence, meaning the height of the blue bars (accuracy) should be as close as possible to the height of the red bars (confidence). When the blue bar is taller than the red bar, it indicates under-confidence. Conversely, if the blue bar is shorter than the red bar, it indicates over-confidence.

- Our method can be seamlessly integrated with the existing post-hoc calibration methods. Extensive experiments demonstrate that our method can outperform the baseline methods, achieving promising performance in the calibration of the GNNs when the graph contains OOD nodes.

- Experimental results further reveal that the learned edge weights are transferable, offering benefits in graph learning across various GNN architectures. Specifically, our optimized graph structure can enhance the performance in tasks such as node classification and OOD detection.

## 2 Related Works

**Neural Network Calibration.** The pursuit of developing a reliable and trustworthy model has captured the attention of researchers, leading to its extension into the realm of graph neural networks. Guo et al. (Guo et al., 2017) first proposed the calibration error to measure the confidence of the results from deep neural networks. Extensive work (Mukhoti et al., 2020; Ghosh et al., 2022; Tao et al., 2023; Wang et al., 2022; 2024; Tang et al., 2024) has been done on the calibration of neural networks. Recent work (Wang et al., 2021) post-processed the logits of the GCN (Kipf & Welling, 2016) model to obtain the calibrated results. Uncertainty estimation (Lakshminarayanan et al., 2017; Malinin & Gales, 2018) also benefits the network calibration by modeling the probability distribution of the predicted labels. Wang et al. (Wang et al., 2022) proposed GCL loss to mitigate the under-confidence issue of GNNs in an end-to-end manner. Besides, GATS (Hsu et al., 2022) is designed to account for the influential factors that affect the calibration of GNN. Fang et al. (Fang et al., 2024) highlighted that the ability of GNNs to distinguish between correct and incorrect predictions is crucial for achieving well-calibrated outcomes. And they propose a simple yet effective approach known as a Discriminative Calibration model for GNNs .Tang et al. (Tang et al., 2024) provided a theoretical insight on the role of nodewise similarity on the calibration of the GNN and proposed a novel calibration framework that takes advantage of the similarity on both global and local levels. Yang et al. (Yang et al., 2024b) highlighted a key limitation in existing GNN calibration methods, which predominantly focus on the highest logit while ignoring the full spectrum of prediction probabilities. To address this issue, they proposed a novel framework called Balanced Calibrated Graph Neural Network (BCGNN) (Yang et al., 2024b). This framework aims to achieve balanced calibration between over-confidence and under-confidence in the GNN predictions, which is supported by the solid theoretical justification. Unlike post-hoc methods that adjust temperature parameters for calibration, Yang et al. (Yang et al., 2024a) attempt to address the calibration

issue from the data centric perspective. And their method aims to lower the calibration error by assigning larger weights to decisive and homophilic edges. Shi et al. (Shi et al., 2023) also investigated the calibration issue of graphs that contain OOD nodes and proposed a new framework called GERDQ. However, there are fundamental differences between our work and GERDQ (Shi et al., 2023). First, we introduce a novel edge iteration approach to better capture the topological structure of the graph. Second, while GERDQ (Shi et al., 2023) also calibrates graphs by reweighting edges, it assigns a fixed value of weight to adjusted edges without considering variations in the topological structure (e.g., the distribution of OOD nodes). In contrast, our method generates fine-grained, topology-aware edge weights that adapt to the graph's structure. Lastly, we design a new reward function to guide the generation of an optimized graph structure, ensuring both improved node classification accuracy and better calibration performance.

**Graph Structure Learning.** Graph Structure Learning (GSL) aims to address graphs with unreliable, low-quality, or noisy structures, such as redundant or incomplete edges, by learning an optimized topology. Up to now, extensive research has been conducted in this field. Wu et al. (Wu et al., 2022) introduced a kernelized Gumbel-Softmax operator for efficiently approximating discrete latent structures among data point and proposed a transformer-based model to learn the optimal topology from node features and labels. To address the lack of robustness and interpretability in existing GSL methods, Zou et al. (Zou et al., 2023) proposed the SE-GSL framework, which can explicitly interpret the hierarchical semantics of graphs, and enhance the robustness of mainstream GNN approaches against noisy and heterophilous structures. To reduce the dependence of GSL methods on label information, Liu et al. (Liu et al., 2022) introduced Structure Bootstrapping contrastive Learning Framework (SUBLIME), a novel unsupervised learning framework that leverages self-supervised contrastive learning to optimize graph structure.

**Reinforcement Learning on Graph.** The rapid development of Reinforcement Learning (RL) in cross-disciplinary domains has motivated scholars to explore novel RL models to address graph-related problems, such as neighborhood detection, information aggregation, and adversarial attacks. GraphNAS (Gao et al., 2019) designs a search space covering sampling functions, aggregation functions, gated functions and searches the graph neural architectures with RL. Policy-GNN (Lai et al., 2020) adaptively determines the number of aggregations for each node via deep Q-learning (Mnih et al., 2013). RL-Explainer (Shan et al., 2021) and GFlowExplainer (Li et al., 2023) adopt off-policy reinforcement learning methods for graph explanation.

**Graph Learning with OOD.** Most graph learning is built on the hypothesis that training and testing data are independent and identically distributed (I.I.D.). Song et al. (Song & Wang, 2022) first proposed graph learning with OOD nodes and developed OODGAT (Song & Wang, 2022) framework to perform both the node classification and OOD nodes detection. The core of the OODGAT (Song & Wang, 2022) is to identify the OOD nodes and reduce the connection between ID nodes and OOD nodes. Another line of work focuses on graph OOD detection. GNNSAGE (Wu et al., 2023) performs OOD node detection by a learning-free energy belief propagation scheme. In GPN (Stadler et al., 2021) OOD nodes detection is completed by the uncertainty estimation. GraphDE (Li et al., 2022), a probabilistic generative framework, can jointly perform graph debiased learning and out-of-distribution nodes detection.

# 3 Preliminary

## 3.1 Problem Formulation

We first present the problem formulation of our study. Consider an attributed graph $\mathcal{G} = \{\mathcal{V}, \mathcal{E}, X\}$ where the finite node set is denoted by $V = \{i | 1 \leq i \leq N\}$, and the edge set is denoted by $\mathcal{E} \subseteq \mathcal{V} \times \mathcal{V}$. $N$ is the total number of the nodes in the graph, and the feature matrix is denoted by $\mathbf{X} \in \mathbb{R}^{N \times d}$ in which $d$ is the length of the feature vector. The structure of the graph $\mathcal{G}$ can be represented by the binary adjacency matrix $\mathbf{A} = \{0, 1\}^{N \times N}$. In graph learning with out-of-distribution (OOD) nodes, The nodes set can be split into an ID node set and an OOD node set $\mathcal{V} = \mathcal{V}_{ID} \cup \mathcal{V}_{OOD}$. The feature of OOD nodes is sampled from a different distribution than that of ID nodes, i.e., $P(X_{OOD}) \neq P(X_{ID})$. The label space for the ID node set is $Y = \{1, 2, \cdots, K\}$, while we assume that the OOD nodes do not fall into any existing category of the ID nodes, and their labels are unknown to us. In semi-supervised graph learning, the ID nodes can be further divided into labeled ID nodes and unlabeled ID nodes, i.e., $\mathcal{V}_{ID} = \mathcal{V}_{ID}^{l} \cup \mathcal{V}_{ID}^{ul}$. The goal of standard

Table 1: Comparison between GCN with original and modified edge weights in terms of node classification accuracy (Acc%)↑ and expected calibration error (ECE%)↓. The experiments are repeated 10 times and the average results are reported. The bold represents the best results.

| Edge weight | Cora | | Citeseer | | PubMed | | CS | | Computers | | Arxiv | |
|---|---|---|---|---|---|---|---|---|---|---|---|---|
| | Acc | ECE | Acc | ECE | Acc | ECE | Acc | ECE | Acc | ECE | Acc | ECE |
| Original | 84.18 | 9.90 | 71.57 | 5.41 | 92.11 | 1.59 | **92.80** | 2.93 | 90.81 | 3.88 | 42.47 | 6.14 |
| Modified | **84.50** | **9.14** | **71.75** | **4.98** | **92.24** | **1.27** | 92.68 | **2.73** | **91.07** | **3.42** | **42.93** | **5.35** |

semi-supervised graph learning is to learn a classifier $f : \mathbf{X}, \mathbf{A} \to \tilde{Y}$ that maps the feature of the nodes and graph structural information to the predicted labels $\tilde{Y}$ of the nodes. As aforementioned, the task becomes more challenging with the presence of unknown OOD nodes. How to rule out the negative impact from the OOD nodes is the key for the semi-supervised graph learning with OOD nodes.

In our study, the expected calibration error (ECE) is considered as a major metric. According to the practice in related work (Guo et al., 2017), the predictions are regrouped into $M$ equally spaced confidence intervals $(B_1, B_2, \cdots, B_M)$ with $B_m = \{i \in \mathcal{V} | \frac{m-1}{M} < \tilde{p}_i \leq \frac{m}{M}\}$ where $\tilde{p}_i$ is the confidence for node $i$. And the expected calibrated error (ECE) can be defined as $\text{ECE} = \sum_{m=1}^{M} \frac{|B_m|}{|\mathcal{V}|} |\text{acc}(B_m) - \text{conf}(B_m)|$, where $\text{acc}(B_m) = \frac{1}{|B_m|} \sum_{i \in B_m} \mathbb{1}(\tilde{y}_i = y_i)$ and $\text{conf}(B_m) = \frac{1}{|B_m|} \sum_{i \in B_m} \tilde{p}_i$.

## 3.2 Deep Reinforcement Learning

Reinforcement learning plays an important role in the decision making process, and one representative method is the Markov Decision Process (MDP). A typical MDP can be formulated as $\mathcal{M} = \{\mathcal{S}, \mathcal{A}, P_\pi, r, \gamma\}, \rho_0$, where $\mathcal{S}$ is the state space, $\mathcal{A}$ is the action space, $P_\pi(s'|s, a)$ is the state-action transition probability, $r$ is the reward function, $r$ is the reward function, $\gamma \in (0, 1)$ is the discount factor ,and $\rho_0$ is the initial state distribution over state space $\mathcal{S}$. The goal of off-policy reinforcement learning is to learn the policy $\pi(a|s)$ that can maximize the discounted cumulative reward $J_\pi = \sum_{t=0}^{\infty} \gamma^t r(s_t, a_t)$ by training on the outcomes produced by a different behavior policy rather than that produced by the target policy. One of the most well-known off-policy method in deep learning is deep Q-learning (Mnih et al., 2013; Van Hasselt et al., 2016). The basic idea of deep Q-learning is to approximate the Q function by deep neural networks, and the policy is obtained from the estimated value of $a = \text{argmax}_a Q(s, a) = \text{argmax}_a \mathbb{E}_{s' \sim \mathcal{S}}(r + \gamma \max_{a'} Q(s', a'))$.

Apart from Q-value based methods that obtain the action implicitly from the Q function, policy gradient methods (Haarnoja et al., 2018; Wang et al., 2017; Cobbe et al., 2021; Barth-Maron et al., 2018; Tkachenko, 2015; Silver et al., 2014b; Mnih et al., 2016) instead aim to learn the policy directly by parameterized function $\pi_\theta(a)$. Similar to deep Q-

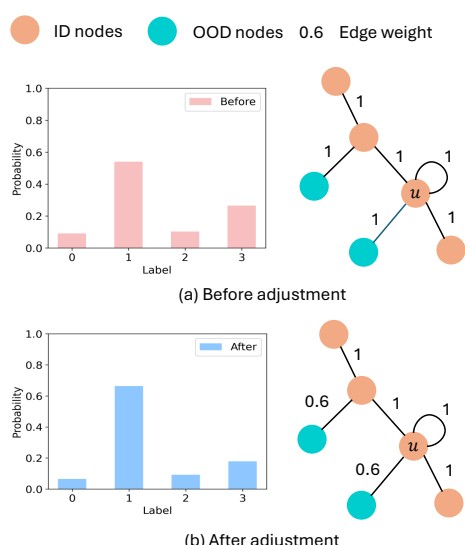

(a) Before adjustment

(b) After adjustment

Figure 3: The change of logit distribution before and after adjustment of edge weight. The result is yielded by GCN on Cora.

learning (Mnih et al., 2013; Van Hasselt et al., 2016), we update the parameter $\theta$ in the policy function to achieve the maximum discounted cumulative reward. Besides, modern off-policy gradient methods (Haarnoja et al., 2018; Wang et al., 2017; Cobbe et al., 2021; Barth-Maron et al., 2018; Tkachenko, 2015) adopt the actor-critic algorithm that models the policy and Q function to achieve better learning efficiency and convergence. The parameter $\theta$ of policy function can be updated according to the Policy Gradient Theorem (Sutton et al., 1999):

$$\nabla_\theta J(\theta) = \mathbb{E}_\pi [\nabla \ln \pi(a|s, \theta) Q_\pi(s, a)]. \tag{1}$$

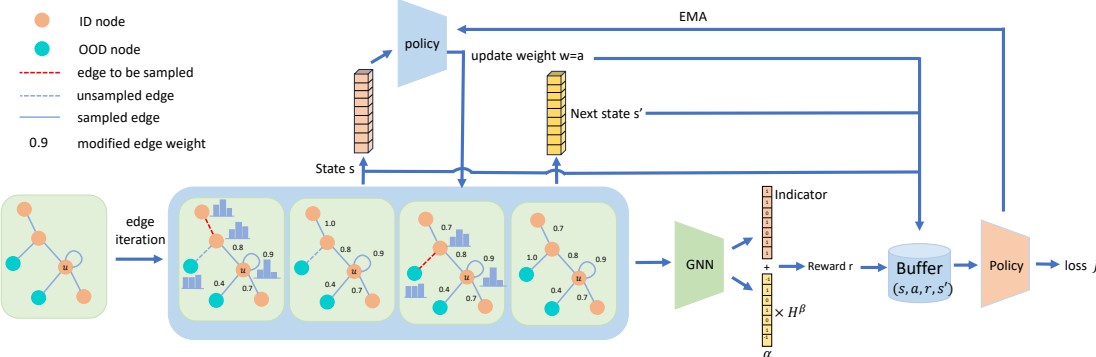

Figure 4: The illustration of our proposed Graph Calibration via Structure Optimization framework. The method consists of four steps. First, we iteratively traverse the adjacent edges from the candidate edge set. In the beginning, only the self-loop edge is taken into consideration. Each time we sample a new edge within the subgraph without replacement according to the discrepancy score and form the state. Second, the adjusted weight would be obtained from the state and assigned to the new sampled edge. Next, reward $r$ is obtained from the GNN with adjusted edge weight and a new state is also formed. Next, the transition tuple is stored in the replay buffer. Finally, we adopt the DDPG method to train our policy function.

## 4 Empirical Study

In this section, we investigate whether the calibration error of GNNs can be reduced by adjusting edge weights in graphs that include out-of-distribution (OOD) nodes. Following previous works (Zhao et al., 2020; Stadler et al., 2021), we divide the nodes into ID and OOD categories and choose GCN (Kipf & Welling, 2016) as the target model. Assuming that the labels and distributions of all nodes are known, we manually modify the weights of edges connected to OOD nodes (e.g., reducing from 1.0 to 0.6). The experiments are evaluated on the six benchmark datasets. The details of the benchmark can be found in Table 2. The results in Table 1 show that reducing the weights of edges linked to OOD nodes can effectively decrease the calibration error while maintaining comparable node classification accuracy relative to using the original edge weights. Our intuition is that adjusting edge weights regularizes the entropy of the node predictions, thereby altering the model's confidence without sacrificing accuracy. To validate this hypothesis, we examine the output distributions of GCN (Kipf & Welling, 2016) on the Cora dataset (Yang et al., 2016) before and after modifying the edge weights. As shown in Fig 3, the results indicate a slight shift in predictive confidence, while the predicted labels remain unchanged. These findings motivate us to design new methods for learning edge weights that improve the calibration of GNNs in the presence of OOD nodes.

## 5 Methodology

In this section, we give an overview of our framework. In this section, we first introduce the formulation of our edge iteration process and the definition of the key elements in our method such as State, Action and Reward. Then we provide details of our training pipeline. Besides, we provide a discussion of our method along with an analysis of the time complexity.

### 5.1 Iterative Edge Sampling and Re-weighting

We first form the candidate nodes set $\mathcal{I}$ which are sampled from node set $\mathcal{V}$. Then the candidate edges for iteration are sampled from adjacent edges of a target node $u \in \mathcal{I}$ and the edge set is denoted as $\mathcal{E}^u = \{e_0^u, e_1^u, \cdots, e_{k-1}^u\}$. Specifically, $e_0^u$ is the self-loop edge for node $u$. During the iteration an edge is sampled from the candidate edge set and the weight is adjusted accordingly. Specifically, At time $t = 0$, we only consider the re-weighting of the self-loop edge $e_0^u$. From time $t - 1$ to $t$, a new edge is chosen from $\mathcal{E}^u$. In our framework the iterative edge sampling and the re-weighting process are formulated as a Markov Decision Process (MDP) and the definitions of state, action, and reward are illustrated as follows.

**State**. The state $s_t \in \mathcal{S}$ at timestamp $t$ in our framework is defined as:

$$s_t = h(s_{t-1}, f_{e_t}), \tag{2}$$

where $f_e$ is the feature of the edge $e$, and $h$ is the function that maps the old state and new edge feature into the new state. We adopt the average of the features of the connecting nodes of the edge $e$ as the edge feature. At time $t = 0$, $s_0 = X_u$. In our study, the moving average method is adopted to generate the state. The state at time $t$ can be formulated as:

$$s_t = \lambda f_{e_t} + (1 - \lambda)s_{t-1}, \tag{3}$$

where $\lambda$ is the hyper-parameter that balances the contribution of new edge features in the state.

In each iteration, a new edge is chosen from the candidate edge set $\mathcal{E}^u$ according to the discrepancy score $\delta$. The discrepancy score $\delta$ measures the correlation between an adjacent edge $e^u$ and the target node $u$. It is defined as:

$$\delta := \frac{1}{2}(\text{KL}(z_1||z_u) + \text{KL}(z_2||z_u)), \tag{4}$$

where $z_1$, $z_2$ and $z_u$ are the logit distribution of the connecting nodes of the edge $e^u$ and target node $u$, respectively. KL denotes the Kullback–Leibler (KL) divergence. Intuitively, the discrepancy score increases when an edge connects to an out-of-distribution (OOD) node whose logit distribution deviates from that of in-distribution (ID) nodes. In each iteration, as the policy function generates new edge weights, the logit distributions of the nodes are updated, and the edge with the lowest discrepancy score is selected from the candidate edge set. Note that edge selection is performed via sampling without replacement, and only the discrepancy scores of unsampled edges are considered for comparison. This selection strategy is motivated by the importance of node similarity in node calibration. By sampling edges based on their discrepancy scores, the method can better isolate the effects of nodes with varying degrees of similarity on the calibration of the target node, thereby facilitating more stable and efficient training convergence.

**Action**. In our method, the action $a \in \mathcal{A}$ we take for each new sampled edge is to adjust its weight. Since in our case, the action space is continuous $\mathcal{A} \subseteq (0, 1]$, we adopt the policy function to generate the adjusted edge weight from the state $s$. At time $t$, the edge weight $w_{e_t}$ for $e_t$ is generated by:

$$w_{e_t} = \pi(s_t|\theta^\pi), \tag{5}$$

where $\pi_\theta$ is the policy function which can be implemented as a neural network with the Sigmoid activation function in the last layer to ensure the output is between 0 and 1.

**Reward**. The reward signal $r$ is designed to encourage the policy function to produce new edge weights to regularize the logit distribution of the target nodes. To determine if the node is over-confident or under-confident, we evaluate the calibration error on the validation nodes and obtain the $\text{acc}(B_m)$ and $\text{conf}(B_m)$ for each bin during training. If the predictive probability of the target node falls into certain bin $m$, then the reward would be defined as

$$r(s, a) = \mathbb{1}(\tilde{y}_i = y_i) + \alpha H_i^\beta, \tag{6}$$

where

$$\begin{cases} \alpha = +1, \beta = 1 & \text{if } \tilde{y}_i = y_i \text{ and } \text{acc}(B_m) < \text{conf}(B_m) \\ \alpha = -1, \beta = 1 & \text{if } \tilde{y}_i = y_i \text{ and } \text{acc}(B_m) > \text{conf}(B_m) \\ \alpha = 0, \beta = 0 & \text{if } \tilde{y}_i = y_i \text{ and } \text{acc}(B_m) = \text{conf}(B_m) \\ \alpha = 1, \beta = 1 & \text{if } \tilde{y}_i \neq y_i \end{cases} \tag{7}$$

$\tilde{y}_i$ is the predicted label for node $i$ generated by the GNN backbone, and $y_i$ is the ground truth label. $H$ denotes the entropy of the target node $u$.

## 5.2 Details of Algorithm

The framework of our proposed method is illustrated in Fig. 4. The framework basically consists of four steps. In this first step, we form the candidate node set $\mathcal{I}$ from the training and validation nodes. For each

candidate node, we iteratively sample the adjacent edges and form the state, as discussed in Sec. 5.1. In step two, the adjusted edge weight is obtained from the policy function $\pi_\theta(s)$. In order to enhance the exploration ability of the policy function in the continuous action space, we reformulate our adjusted edge weight as:

$$w^*_{e_t} = \pi(s_t|\theta^\pi) + \epsilon, \tag{8}$$

where $\epsilon$ is the noise following a uniform distribution and the upper bound is determined by $\epsilon_0(1+\frac{t}{T})^{-d}$, where $\epsilon_0$ is the initial noise. $T$ is the total number of iterations. $d > 0$ is the decay rate. In the next step we obtain the reward $r$ from the GNN backbone according to Eq. 6 and the tuple of transition $(s_t, a_t, r_t, s_{t+1})$ is stored in the replay buffer $\mathcal{B}$. In the final step, we adopt the Deep Deterministic Policy Gradient (DDPG) (Lillicrap et al., 2016) method to train our policy function. Similar to deep Q-learning (Mnih et al., 2013), the objective of critic network $Q(s_t, a_t|\theta^Q)$ is to approximate the discounted cumulative reward from the state-action pair by minimizing the loss:

$$L(\theta^Q) = \mathbb{E}_{s_t \sim \mathcal{S}, a_t \sim \mathcal{A}}[(Q(s_t, a_t)|\theta^Q] - y_t)^2], \tag{9}$$

where $y_t$ can be derived from the Bellman equation (Sutton & Barto, 2018) $y_t = r(s_t, a_t) + \gamma Q(s_{t+1}, \pi(s_{t+1}|\theta^\pi)|\theta^Q)$ . Since our policy function yields the continuous edge weight deterministically from the state, the parameter of policy can be updated according to the Deterministic Policy Gradient Theorem (Silver et al., 2014a; Lillicrap et al., 2016):

$$\nabla_{\theta^\pi} J = \mathbb{E}_{s_t}[\nabla_a Q(s, a|\theta^Q)_{s=s_t, a=\pi(s_t|\theta^\pi)} \nabla_{\theta^\pi} \pi(s|\theta^\pi)_{s=s_t}]$$
$$\approx \frac{1}{N} \sum_i (\nabla_a Q(s, a|\theta^Q)_{s=s_i, a=\pi(s_i|\theta^\pi)} \nabla_{\theta^\pi} \pi(s|\theta^\pi)_{s=s_i}). \tag{10}$$

The detailed procedures of our proposed method are summarized in Algorithm 1 in the Appendix.

### 5.3 Time Complexity

Our framework consists of a GNN and actor/critic network. Suppose the $L$ is the number of layers in GCN, $|E|$ is the total number of edges, $N$ is the total number of nodes, $d$ is the dimension of the features, $|\mathcal{E}^u|$ is the number of edges in the candidate edge set. $|\mathcal{I}|$ is the number of the target nodes. The time complexity for GNN and actor/critic networks are $O(L|E||\mathcal{E}^u|d + LN|\mathcal{E}^u|d^2)$ and $O(|\mathcal{I}||\mathcal{E}^u|d)$, respectively. Thus, the total time complexity is $O(|\mathcal{E}^u|(L|E| + |\mathcal{I}|)d + LN|\mathcal{E}^u|d^2)$. The major factors that influence computational efficiency in our approach are the number of target nodes and the number of adjacent edges for iterations. As the graph size increases, the number of nodes and adjacent edges also grows, raising the computational cost of our method. To address this issue, we limit the number of target nodes and adjacent edges for each iteration. For larger graphs like OGB-Arxiv (Hu et al., 2020), we select up to 48 target nodes and 64 adjacent edges per node. Additionally, to further improve computational efficiency, we

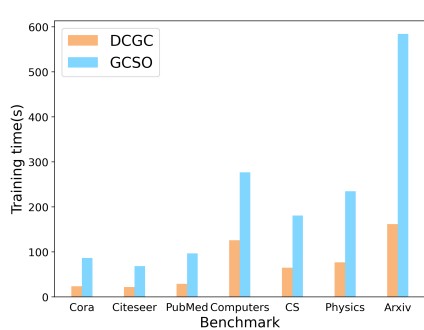

Figure 5: The actual training time of the baseline (DCGC) our method on different benchmarks. The evaluation is conducted on an NVIDIA RTX A5000.

process edge iterations for all target nodes in batches. As a result, our method is scalable to large graphs and it can maintain a reasonable computational cost even as graph size increases. During the inference time, the modified edge weights would be yielded by our lightweight policy function in batch. Fig 5 illustrates the actual training time of the data-centric method DCGC (Yang et al., 2024a) and our proposed method on different benchmarks. The results suggest that, although our method requires more training time than the baseline, it maintains a reasonable computational cost even for large graphs.

### 5.4 Discussion

In our method, we design a specialized reward signal to refine the graph structure, which could implicitly regularize the logit distribution of the GNNs. When the target node is correctly predicted and the corresponding accuracy is lower than the confidence, the reward is $r = 1 + H$. An increasing reward enlarges the

Table 2: The statistics of datasets

| Dataset | ID classes | OOD classes | #Nodes | #Edges | #Features |
|---|---|---|---|---|---|
| Cora | [0 - 4] | [5 - 6] | 2,708 | 10,556 | 1,433 |
| Citeseer | [0 - 3] | [4 - 5] | 3,327 | 9,104 | 3,703 |
| PubMed | [0 - 1] | [2] | 19,717 | 88,648 | 500 |
| Chameleon | [0 - 2] | [3 - 4] | 2,277 | 36,101 | 2,325 |
| Coauthor-CS | [0 - 10] | [11 - 14] | 18,333 | 163,788 | 6,805 |
| Coauthor-Physics | [0 - 2] | [3] | 34,493 | 495,924 | 8,415 |
| Amazon-Computers | [0 - 6] | [7 - 9] | 13,752 | 491,722 | 767 |
| OGB-Arxiv | [0 - 29] | [30 - 39] | 169,343 | 1,166,243 | 128 |

entropy, making the logit distribution less concentrated (e.g., reducing confidence). When the corresponding accuracy exceeds the confidence, the reward $r = 1 - H$ reduces the entropy, making the output distribution more concentrated (e.g., increasing confidence). If the target node is not correctly predicted. The reward $r = H$ simply makes the model deviate from the current output distribution.

Next, we discuss the generalization of our method. Our policy function is trained on selected edges within a graph. Since there is no distribution shift within a single graph, our method can generalize to other edges in the same graph. However, due to the differences in topological structures across graphs, the policy functions for different graphs need to be trained separately. Another advantage of our method is that the modified graph structure can be leveraged by other calibration frameworks (e.g., CaGCN (Wang et al., 2021) and GATS (Hsu et al., 2022)) to achieve better calibration results.

Besides, earlier work (Zhang et al., 2020) suggests three desired properties for calibration methods: **accuracy-preserving**, **data-efficient**, and **expressive**. Our method can fulfill these properties. Since the modified graph structure may affect the quality of the learned representations, and thereby influence the performance on downstream tasks such as node classification, we incorporate an indicator function in the reward signal. This component empirically encourages the preservation of classification accuracy on the ID nodes during graph structure optimization. Besides, our method is also data-efficient. In our method we adopt GCN (Kipf & Welling, 2016) as the GNN backbone and lightweight MLPs for the actor and critic models, respectively. At last, our method is expressive, as it could produce a fine-grained, topology-aware weight for each edge.

Finally, our work aligns with a prior study (Shi et al., 2023) that also addresses the calibration issue of graph neural networks (GNNs) in out-of-distribution (OOD) scenarios. Although both works utilize reinforcement learning, there are significant differences between the two studies. First, the previous work lacks a solid empirical foundation and is driven by more intuitive motivations. In contrast, our research identifies the presence of both under-confidence and over-confidence in GNNs with the presence of OOD nodes, an observation not made in the prior study which led us to develop a different approach. Furthermore, compared to the previous work (Shi et al., 2023), we develop a new edge iteration process to capture the topological information of the graph. Furthermore, we propose a novel reward signal to generate fine-grained, topology-aware edge weights for the adjusted graph structure. In contrast, the previous work (Shi et al., 2023) produces fixed edge weights without considering topological information, such as the distribution of OOD nodes.

## 6 Experiments

In this section, we first introduce the experimental settings. Then we show the main results of the experiment as well as the visualization of the reliability diagrams and the distribution of the modified edge weights. The results of ablation study and case study can be found in Appendix.

Table 3: Comparison between our proposed method and other baselines in terms of node classification accuracy (Acc%)↑ and expected calibration error (ECE%)↓ on Cora, Citeseer and PubMed. The experiments are repeated 10 times and the average results and standard deviation are reported. Note that the primary focus of our study is the ECE performance of the methods.

| Methods | Cora | | Citeseer | | PubMed | | Chameleon | |
|---|---|---|---|---|---|---|---|---|
| | Acc | ECE | Acc | ECE | Acc | ECE | Acc | ECE |
| GCN (Kipf & Welling, 2016) | $84.18 \pm 0.28$ | $9.90 \pm 0.61$ | $71.57 \pm 0.73$ | $5.41 \pm 1.51$ | $92.11 \pm 0.17$ | $1.59 \pm 0.56$ | $47.27 \pm 1.27$ | $14.81 \pm 2.14$ |
| GCL (Wang et al., 2022) | $84.19 \pm 0.25$ | $10.05 \pm 0.63$ | $71.91 \pm 0.96$ | $6.07 \pm 2.03$ | $92.14 \pm 0.14$ | $1.56 \pm 0.25$ | $45.81 \pm 0.72$ | $13.98 \pm 1.79$ |
| OODGAT (Song & Wang, 2022) | $83.17 \pm 1.34$ | $13.96 \pm 3.87$ | $61.95 \pm 0.78$ | $8.52 \pm 2.08$ | $87.44 \pm 0.91$ | $4.64 \pm 1.28$ | $39.74 \pm 4.57$ | $11.06 \pm 5.90$ |
| HyperU-GCN (Yang et al., 2022) | $81.88 \pm 1.09$ | $8.40 \pm 7.75$ | $71.27 \pm 1.39$ | $19.69 \pm 13.74$ | $92.35 \pm 0.48$ | $3.24 \pm 0.77$ | $47.36 \pm 2.29$ | $15.22 \pm 9.65$ |
| CaGCN (Wang et al., 2021) | $84.14 \pm 0.35$ | $3.85 \pm 1.05$ | $71.57 \pm 0.73$ | $4.27 \pm 0.62$ | $92.11 \pm 0.17$ | $3.09 \pm 0.20$ | $47.27 \pm 1.27$ | $14.15 \pm 1.24$ |
| GATS (Hsu et al., 2022) | $83.49 \pm 0.31$ | $2.81 \pm 0.82$ | $72.04 \pm 0.46$ | $5.05 \pm 1.86$ | $92.56 \pm 0.24$ | $2.27 \pm 0.39$ | $46.76 \pm 2.73$ | $11.18 \pm 3.62$ |
| GERDQ (Shi et al., 2023) | $83.67 \pm 0.48$ | $9.54 \pm 0.50$ | $69.98 \pm 0.55$ | $4.36 \pm 0.92$ | $92.14 \pm 0.20$ | $1.60 \pm 0.59$ | $46.87 \pm 1.54$ | $14.22 \pm 1.65$ |
| DCGC (Yang et al., 2024a) | $83.91 \pm 0.25$ | $10.44 \pm 0.76$ | $65.02 \pm 0.65$ | $4.62 \pm 1.07$ | $92.26 \pm 0.16$ | $2.43 \pm 0.44$ | $\mathbf{47.60} \pm 3.22$ | $13.28 \pm 4.42$ |
| GCSO (Ours) | $\mathbf{84.95} \pm 0.18$ | $9.22 \pm 0.49$ | $71.80 \pm 0.70$ | $4.55 \pm 1.63$ | $92.16 \pm 0.16$ | $\mathbf{1.49} \pm 0.20$ | $46.81 \pm 1.03$ | $13.03 \pm 1.59$ |
| GCSO+CaGCN (Ours) | $84.28 \pm 0.27$ | $\mathbf{2.55} \pm 0.45$ | $71.82 \pm 0.68$ | $\mathbf{4.15} \pm 0.48$ | $92.24 \pm 0.26$ | $2.80 \pm 0.19$ | $47.28 \pm 1.21$ | $13.78 \pm 0.82$ |
| GCSO+GATS (Ours) | $84.20 \pm 0.31$ | $2.63 \pm 0.46$ | $\mathbf{72.24} \pm 0.90$ | $4.20 \pm 0.48$ | $\mathbf{92.69} \pm 0.27$ | $2.13 \pm 0.34$ | $46.80 \pm 1.95$ | $\mathbf{10.07} \pm 2.07$ |

Table 4: Comparison between our proposed method and other baselines on Photo, Computers and Arxiv in terms of node classification accuracy (Acc%)↑ and expected calibration error (ECE%)↓. The experiments are repeated 10 times and the average results and standard deviation are reported. Note that the primary focus of our study is the ECE performance of the methods.

| Methods | Coauthor-CS | | Coauthor-Physics | | Amazon-Computers | | OGB-Arxiv | |
|---|---|---|---|---|---|---|---|---|
| | Acc | ECE | Acc | ECE | Acc | ECE | Acc | ECE |
| GCN (Kipf & Welling, 2016) | $91.96 \pm 0.72$ | $2.57 \pm 0.13$ | $97.02 \pm 0.21$ | $1.27 \pm 0.14$ | $90.81 \pm 0.53$ | $3.02 \pm 0.68$ | $42.47 \pm 0.67$ | $6.14 \pm 0.79$ |
| GCL (Wang et al., 2022) | $91.83 \pm 0.41$ | $2.91 \pm 0.29$ | $97.04 \pm 0.21$ | $1.31 \pm 0.12$ | $90.65 \pm 1.04$ | $3.40 \pm 0.59$ | $42.53 \pm 0.63$ | $6.41 \pm 0.79$ |
| OODGAT (Song & Wang, 2022) | $90.63 \pm 0.35$ | $4.16 \pm 0.60$ | $93.79 \pm 0.52$ | $3.44 \pm 0.38$ | $90.29 \pm 1.02$ | $4.84 \pm 1.01$ | $42.11 \pm 1.19$ | $11.64 \pm 0.74$ |
| HyperU-GCN (Yang et al., 2022) | $90.74 \pm 1.03$ | $2.94 \pm 0.84$ | $96.37 \pm 0.57$ | $1.86 \pm 0.70$ | $90.17 \pm 1.37$ | $5.82 \pm 1.10$ | $36.72 \pm 0.65$ | $13.23 \pm 1.58$ |
| CaGCN (Wang et al., 2021) | $89.79 \pm 0.40$ | $4.34 \pm 0.40$ | $97.04 \pm 0.20$ | $1.09 \pm 0.13$ | $88.67 \pm 0.38$ | $2.82 \pm 0.17$ | $41.99 \pm 0.75$ | $4.42 \pm 0.35$ |
| GATS (Hsu et al., 2022) | $89.28 \pm 0.47$ | $4.14 \pm 0.40$ | $96.80 \pm 0.34$ | $1.21 \pm 0.21$ | $88.07 \pm 0.43$ | $3.59 \pm 0.68$ | $42.07 \pm 0.79$ | $4.84 \pm 0.36$ |
| GERDQ (Shi et al., 2023) | $\mathbf{92.36} \pm 0.50$ | $3.12 \pm 0.25$ | $97.05 \pm 0.23$ | $1.38 \pm 0.23$ | $90.52 \pm 0.42$ | $2.52 \pm 0.49$ | $43.58 \pm 0.60$ | $4.70 \pm 0.48$ |
| DCGC (Yang et al., 2024a) | $92.03 \pm 0.30$ | $3.36 \pm 0.20$ | $96.97 \pm 0.17$ | $1.52 \pm 0.17$ | $90.84 \pm 0.66$ | $2.49 \pm 0.43$ | $43.62 \pm 0.54$ | $4.85 \pm 0.39$ |
| GCSO (Ours) | $91.96 \pm 0.25$ | $\mathbf{2.47} \pm 0.13$ | $\mathbf{97.08} \pm 0.21$ | $1.24 \pm 0.12$ | $\mathbf{90.86} \pm 0.45$ | $\mathbf{2.44} \pm 0.28$ | $\mathbf{43.64} \pm 0.93$ | $4.35 \pm 0.49$ |
| GCSO+CaGCN (Ours) | $89.75 \pm 0.45$ | $4.17 \pm 0.16$ | $97.02 \pm 0.21$ | $\mathbf{1.06} \pm 0.06$ | $88.50 \pm 0.30$ | $2.64 \pm 0.29$ | $42.59 \pm 0.51$ | $\mathbf{4.16} \pm 0.39$ |
| GCSO+GATS (Ours) | $89.28 \pm 0.46$ | $3.92 \pm 0.38$ | $96.78 \pm 0.33$ | $1.15 \pm 0.09$ | $88.49 \pm 0.71$ | $3.33 \pm 0.34$ | $42.89 \pm 0.71$ | $4.37 \pm 0.64$ |

## 6.1 Experimental Settings

In the experiments, we perform the semi-supervised node classification task and compare the performance of our framework with the baseline methods on eight benchmark datasets. The ablation study and case study can be found in Appendix.

**Datasets**. We adopt eight public benchmark datasets, including Cora, Citeseer, PubMed (Yang et al., 2016), Chameleon (Rozemberczki et al., 2021), Coauthor-CS, Coauthor-Physics, Amazon-Computers (Shchur et al., 2018) and OGB-Arxiv (Hu et al., 2020), for evaluating our method and baselines. Among the dataset, the Chameleon (Rozemberczki et al., 2021)is a heterophilous graph while others are homophilous graphs. We adhere to the train/validation/test splits provided by previous work (Hsu et al., 2022; Yang et al., 2016; Shchur et al., 2018). To formulate the graph learning with the OOD nodes setting, we manually split the nodes into ID nodes and OOD nodes according to the routine from the previous work (Gal & Ghahramani, 2016; Song & Wang, 2022; Stadler et al., 2021). For instance, Cora (Yang et al., 2016) has 7 classes and the nodes from the last 2 classes would be regarded as OOD nodes which are marked out in the training and validation data. The rest would be ID nodes. More details of the datasets are illustrated in Table 2.

**Baselines**. The baselines include GCN (Kipf & Welling, 2016), HyperU-GCN (Yang et al., 2022), CaGCN (Wang et al., 2021), GATS (Hsu et al., 2022), GCL (Wang et al., 2022), OODGAT (Song & Wang, 2022), GERDQ (Shi et al., 2023) and DCGC (Yang et al., 2024a). More details about the baseline methods can be found in Appendix.

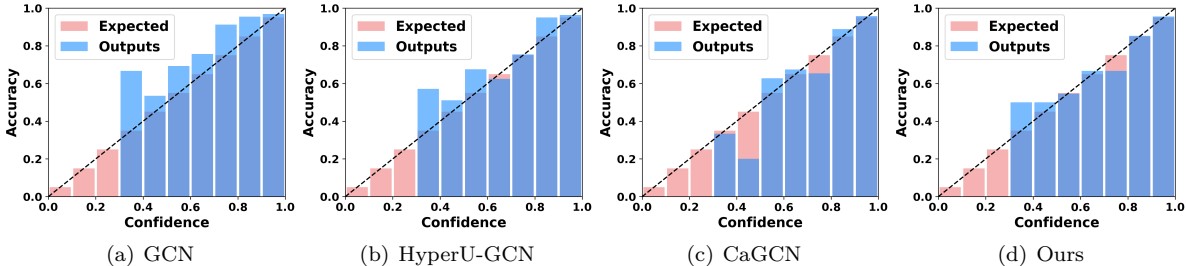

Figure 6: Reliability diagrams of different methods on Cora with OOD nodes. Well-calibrated results would have closer alignment with the expected results along the diagonal line. The results suggest that the calibration issue is different and complicated on different datasets.

Table 5: Comparison between our proposed method and baselines in terms of node classification accuracy(Acc%)↑ and expected calibration error(ECE%)↓ on Cora with different OOD configurations.The experiments are repeated 10 times and the average results and standard deviation are reported.

| Methods | Config 1 | | Config 2 | | Config 3 | |
|---|---|---|---|---|---|---|
| | Acc | ECE | Acc | ECE | Acc | ECE |
| GCN (Kipf & Welling, 2016) | $81.91 \pm 0.57$ | $9.88 \pm 0.71$ | $84.85 \pm 0.22$ | $9.42 \pm 0.75$ | $84.18 \pm 0.28$ | $9.90 \pm 0.61$ |
| CaGCN (Wang et al., 2021) | $81.88 \pm 0.50$ | $4.10 \pm 0.44$ | $84.82 \pm 0.26$ | $3.39 \pm 0.57$ | $84.14 \pm 0.35$ | $3.85 \pm 1.05$ |
| GATS (Hsu et al., 2022) | $\mathbf{82.61} \pm 0.71$ | $4.11 \pm 0.62$ | $85.42 \pm 0.54$ | $3.03 \pm 1.13$ | $83.49 \pm 0.31$ | $2.81 \pm 0.82$ |
| GERDQ (Shi et al., 2023) | $82.02 \pm 0.59$ | $9.46 \pm 0.60$ | $84.53 \pm 0.46$ | $9.25 \pm 0.35$ | $83.67 \pm 0.48$ | $9.54 \pm 0.50$ |
| DCGC (Yang et al., 2024a) | $80.86 \pm 0.18$ | $8.62 \pm 0.70$ | $83.16 \pm 0.44$ | $8.21 \pm 0.48$ | $82.46 \pm 0.26$ | $8.57 \pm 0.62$ |
| GCSO (Ours) | $81.66 \pm 0.56$ | $9.27 \pm 0.43$ | $84.77 \pm 0.23$ | $9.11 \pm 0.17$ | $\mathbf{84.95} \pm 0.18$ | $9.22 \pm 0.49$ |
| GCSO+CaGCN (Ours) | $82.14 \pm 0.42$ | $3.75 \pm 0.38$ | $84.80 \pm 0.28$ | $2.83 \pm 0.38$ | $84.28 \pm 0.27$ | $\mathbf{2.55} \pm 0.45$ |
| GCSO+GATS (Ours) | $81.89 \pm 0.54$ | $\mathbf{2.49} \pm 0.33$ | $\mathbf{85.46} \pm 0.78$ | $\mathbf{2.78} \pm 0.49$ | $84.20 \pm 0.31$ | $2.63 \pm 0.46$ |

**Metrics**. In our experiments, we adopt the expected calibration error (ECE) (Guo et al., 2017) as our major metric. The lower value of ECE means the better reliability of the prediction results from GNN models. Besides, we also report the node classification accuracy.

**Implementation Details**. In our method, we adopt GCN (Kipf & Welling, 2016) as our GNN backbone. The hyper-parameters of GCN are the same as the corresponding baselines. The learning rate is 1e-2 and weight decay is 5e-4. The hidden dimension is 64. The Actor and Critic in our framework are implemented as a three-layered MLP with the dimension of hidden layers 256 and 16, respectively. More details about the implementation details can be found in Appendix.

## 6.2 Main Results

Table 3 and Table 4 show the performance of our proposed method and the baselines on the benchmarks. The results show that the ordinary GNN models such as GCN (Kipf & Welling, 2016) would yield large calibration errors. For instance, the ECE can reach an average 9.90% on Cora (Yang et al., 2016). Besides, the results also suggest that the methods aimed at the calibration of GNNs can basically improve the calibration performance on GCN (Kipf & Welling, 2016). However, it could fail on some benchmark datasets. For instance, Although CaGCN (Wang et al., 2021) can achieve the low calibration error on Cora (Yang et al., 2016) and Citeseer (Yang et al., 2016), it still results in poorer calibration on PubMed (Yang et al., 2016) and Coauthor-CS (Shchur et al., 2018) than that of GCN (Kipf & Welling, 2016). The cause of the phenomenon can be attributed to the compromised homophily of the graph and make the regularization term in CaGCN (Shchur et al., 2018) less effective on these dataset. GCL (Wang et al., 2022) is also less effective on some datasets. OODGAT (Song & Wang, 2022) can identify the potential OOD nodes during the training and reduce the connection between the ID and OOD nodes by lowering the corresponding edge weights. However, our experimental results show that it would still suffer large calibration errors on some benchmark datasets. Additionally, our experimental results reveal that GERDQ (Shi et al., 2023) and DCGC (Yang et al., 2024a) effectively reduce the calibration error across various benchmarks. This

Table 6: Comparison between our proposed method and baselines in terms of node classification accuracy(Acc%)↑ and expected calibration error(ECE%)↓ on PubMed with different OOD configurations.The experiments are repeated 10 times and the average results and standard deviation are reported.

| Methods | Config 1 | | Config 2 | | Config 3 | |
|---|---|---|---|---|---|---|
| | Acc | ECE | Acc | ECE | Acc | ECE |
| GCN (Kipf & Welling, 2016) | 83.65 ± 0.27 | 3.95 ± 0.76 | 90.32 ± 0.30 | 1.77 ± 0.74 | 92.11 ± 0.17 | 1.59 ± 0.56 |
| CaGCN (Wang et al., 2021) | 83.67 ± 0.28 | 5.07 ± 0.50 | 90.32 ± 0.31 | 1.64 ± 0.44 | 92.11 ± 0.17 | 3.09 ± 0.20 |
| GATS (Hsu et al., 2022) | 83.47 ± 0.32 | 3.67 ± 0.58 | 89.81 ± 0.30 | 1.47 ± 0.60 | 92.56 ± 0.24 | 2.27 ± 0.39 |
| GERDQ (Shi et al., 2023) | 83.65 ± 0.27 | 3.69 ± 0.57 | 90.32 ± 0.30 | 1.62 ± 0.61 | 92.14 ± 0.20 | 1.60 ± 0.59 |
| DCGC (Yang et al., 2024a) | 83.39 ± 0.41 | 3.68 ± 0.70 | 89.88 ± 0.29 | 2.41 ± 0.33 | 92.66 ± 0.16 | 2.43 ± 0.44 |
| GCSO (Ours) | **83.73** ± 0.28 | 3.68 ± 0.51 | **90.39** ± 0.32 | 1.36 ± 0.43 | 92.16 ± 0.16 | **1.49** ± 0.20 |
| GCSO+CaGCN (Ours) | 83.70 ± 0.26 | 4.86 ± 0.55 | **90.39** ± 0.30 | 1.60 ± 0.21 | 92.24 ± 0.26 | 2.80 ± 0.19 |
| GCSO+GATS (Ours) | 83.52 ± 0.45 | **3.42** ± 0.39 | 89.69 ± 0.40 | **1.30** ± 0.19 | **92.69** ± 0.27 | 2.13 ± 0.34 |

Table 7: Comparison between our proposed method and baselines in terms of node classification accuracy(Acc%)↑ and expected calibration error(ECE%)↓ on Computers with different OOD configurations.The experiments are repeated 10 times and the average results and standard deviation are reported.

| Methods | Config 1 | | Config 2 | | Config 3 | |
|---|---|---|---|---|---|---|
| | Acc | ECE | Acc | ECE | Acc | ECE |
| GCN (Kipf & Welling, 2016) | 87.24 ± 0.58 | 2.74 ± 0.40 | **93.79** ± 0.39 | 2.61 ± 0.23 | 90.81 ± 0.53 | 3.02 ± 0.68 |
| CaGCN (Wang et al., 2021) | 88.14 ± 0.24 | 4.41 ± 0.54 | 92.25 ± 0.42 | 3.16 ± 0.35 | 88.67 ± 0.38 | 2.82 ± 0.17 |
| GATS (Hsu et al., 2022) | 87.50 ± 0.77 | 3.21 ± 0.53 | 92.25 ± 0.49 | 2.75 ± 0.41 | 88.07 ± 0.43 | 3.59 ± 0.68 |
| GERDQ (Shi et al., 2023) | 87.08 ± 0.54 | 2.60 ± 0.42 | 93.41 ± 0.65 | 2.76 ± 0.34 | 90.52 ± 0.42 | 2.52 ± 0.49 |
| DCGC (Yang et al., 2024a) | 87.90 ± 0.36 | 3.32 ± 0.71 | 93.64 ± 0.55 | 2.66 ± 0.41 | 90.84 ± 0.66 | 2.49 ± 0.43 |
| GCSO (Ours) | 87.17 ± 0.41 | **2.56** ± 0.35 | 93.16 ± 0.52 | 2.58 ± 0.16 | **90.86** ± 0.45 | **2.44** ± 0.28 |
| GCSO+CaGCN (Ours) | **88.30** ± 0.56 | 4.03 ± 0.45 | 92.56 ± 0.48 | 2.84 ± 0.19 | 88.50 ± 0.30 | 2.64 ± 0.29 |
| GCSO+GATS (Ours) | 87.65 ± 0.75 | 3.07 ± 0.30 | 92.17 ± 0.35 | **2.40** ± 0.21 | 88.49 ± 0.71 | 3.33 ± 0.34 |

further validates that refining the graph topology is beneficial for lowering the calibration error, particularly in graphs with the presence of OOD nodes.

The experimental results suggest that our proposed GCSO method is effective in calibrating GNNs and achieves better ECE performance compared to GERDQ (Shi et al., 2023) and DCGC (Yang et al., 2024a) on most datasets. DCGC (Yang et al., 2024a) can generate an adjusted graph structure to reduce the calibration error. However, this method does not account for the presence of OOD nodes. GERDQ (Shi et al., 2023) also employed adjusted weights to calibrate GNN results. However, its adjustment is guided solely by the accuracy signal, which limits its calibration performance. In contrast, our approach employs a specialized reward signal to dynamically assess the impact of adjusted edge weights on target nodes, generating an optimal, topology-aware graph structure that regularizes the predictive confidence of GNNs. When integrated with existing graph calibration methods, our approach can further enhance calibration performance. For instance, GCSO+CaGCN can approximately achieve the ECE of 2.55% and 1,06% on Cora (Yang et al., 2016) and Coauthor-Physics (Shchur et al., 2018).

To further assess the generalizability and robustness of our proposed method, we expanded our experiments by evaluating it on datasets with different OOD node distributions. Specifically, we conducted evaluations on Cora, PubMed, and Amazon-Computers, each configured with distinct OOD settings. The

Table 8: The different OOD configuration of Cora, PubMed and Amazon_Computers.

| Dataset | Config 1 | | Config 2 | | Config 3 | |
|---|---|---|---|---|---|---|
| | ID classes | OOD classes | ID classes | OOD classes | ID classes | OOD classes |
| Cora | [0,3-6] | [1,2] | [0-2,5,6] | [3,4] | [0-4] | [5,6] |
| PubMed | [1,2] | [0] | [0,2] | [1] | [0,1] | [2] |
| Computers | [0,4-9] | [1,2,3] | [0-3,7-9] | [4,5,6] | [0-6] | [7,8,9] |

OOD configurations are detailed in Table 8, and the corresponding results are reported in Tables 5, Table 6 and Table 7. The results suggest that conventional methods such as CaGCN (Wang et al., 2021) struggle to calibrate GNNs under varying OOD configurations. For example, on the Amazon-Computers dataset,

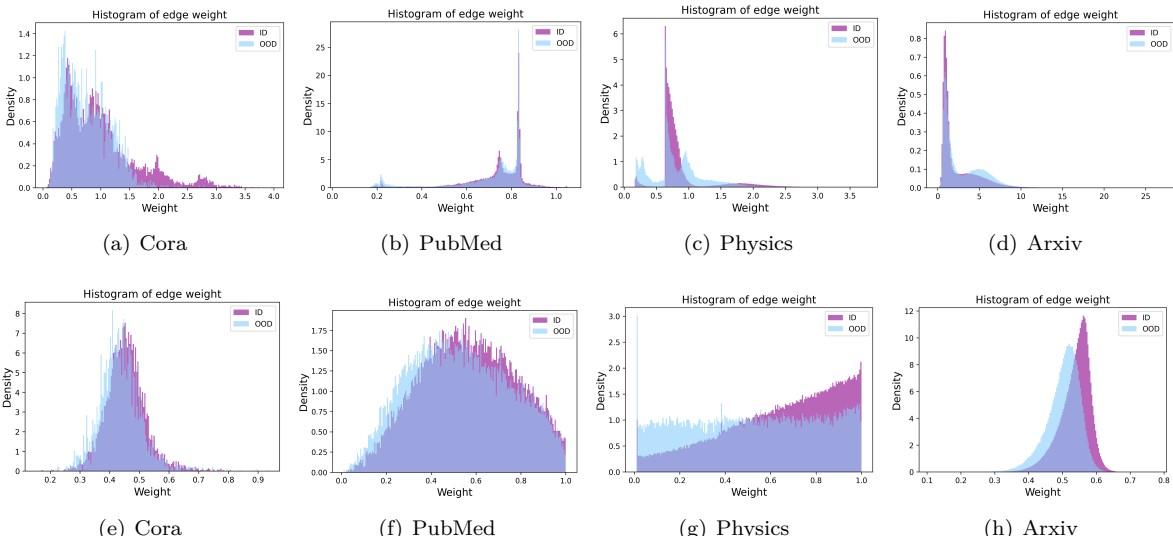

Figure 7: The distribution of edge weights connecting to ID nodes and OOD nodes from DCGC ((a)-(d)) and our proposed method ((e)-(h)) on various datasets. Compared to DCGC, the edge weights generated by our method exhibit a clear distribution shift between ID and OOD nodes.

CaGCN (Wang et al., 2021) yields higher calibration errors in both Configuration 1 (4.41%) and Configuration 2 (3.16%) compared to the baseline GCN (2.74% and 2.61%, respectively). This failure may stem from the method's disregard for graph topology, particularly the distribution of OOD nodes, when adjusting the output logits. In contrast, our proposed method consistently improves calibration across different OOD settings. The improvements are more pronounced on large graphs (e.g., PubMed and Amazon-Computers) than on smaller graphs (e.g., Cora). These results across three datasets empirically demonstrate the generalizability and robustness of our approach under diverse OOD distributions.

### 6.3 Visualization

To better visualize the ECE, the reliability diagrams for our method and the baselines On Cora (Yang et al., 2016) are illustrated in Fig. 6. Well-calibrated results are supposed to have closer alignment with the expected results along the diagonal line. Fig. 6 demonstrates the better alignment of our method to the diagonal line than that of other baselines, which is consistent with our experimental results. The experiments suggest that both two terms are an indispensable part of the reward signal in our framework.

We also visualize the modified graph structures of both DCGC (Yang et al., 2024a) and our proposed method on various datasets. The histogram in Fig. 7 illustrates the distribution of edge weights connecting ID and OOD nodes, respectively. Since DCGC (Yang et al., 2024a) doesn't explicitly account for OOD nodes, the distribution of edge weights connecting to ID and OOD nodes shows significant overlap. In contrast, our method generates topology-aware edge weights. Fig. 7 demonstrates a clear distribution shift between the edge weights connecting to ID and OOD nodes.

### 6.4 Ablation Study

In our framework, the reward consists of two terms: indicator function and entropy regularization term. To investigate the contribution of each term in the reward, we conduct an experiment in which only one term is available in the reward. The results are shown in Table 9 in Appendix. The indicator function is designed to encourage the model to maintain comparable accuracy. Without it, our method tends to suffer a drop in accuracy on in-distribution (ID) nodes, along with increased calibration error. Similarly, removing the entropy regularization term from the reward function reduces the effectiveness of our approach in calibrating graph neural networks.

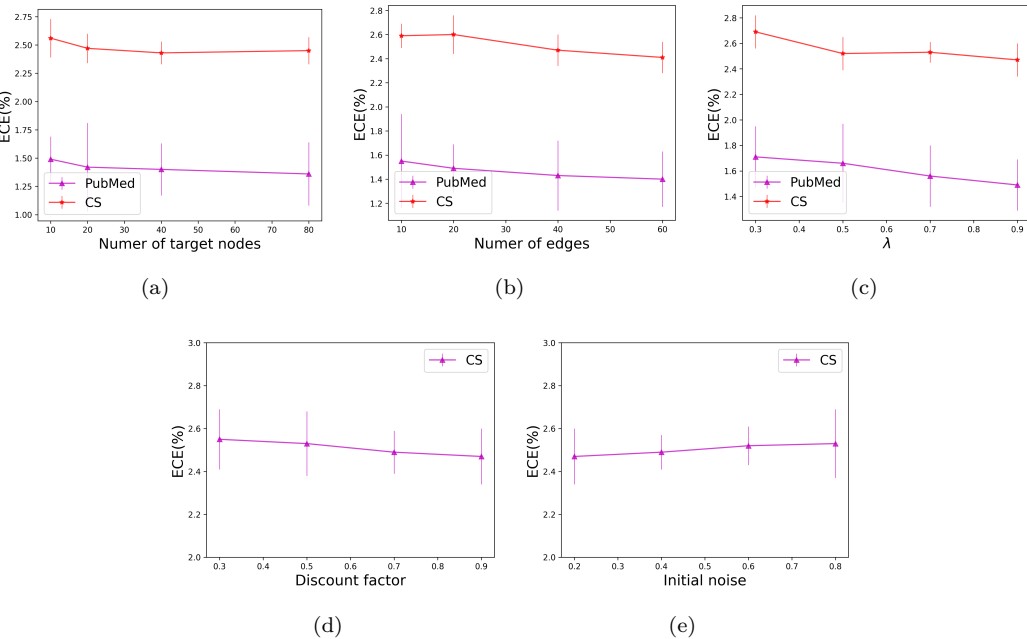

Figure 8: Performance of our method on node classification and calibration on PubMed and Coauthor-CS (a) with varying numbers of labelled nodes, (b) edges and (c) value of $\lambda$. Performance of our method on node classification and calibration on Coauthor-CS with varying value of (d) discounter factor $\gamma$ and (e) initial noise $\epsilon_0$.

Next, we evaluate the impact of the number of sampled target nodes and edges on the performance of our method, we conducted additional experiments on the PubMed and Coauthor-CS datasets. In the first experiment, we fixed the number of sampled edges to 20 for PubMed and 40 for Coauthor-CS, and evaluated the performance using 10, 20, 40, and 80 sampled nodes. In the second experiment, we fixed the number of sampled nodes to 10 for PubMed and 20 for Coauthor-CS, and varied the number of sampled edges among 10, 20, 40, and 60. The corresponding results are presented in Fig. 8(a) and Fig. 8(b), respectively. These results indicate that increasing the number of sampled nodes and edges generally improves calibration performance. However, beyond a certain point, the improvements become marginal while the computational cost increases.

We also conducted an evaluation to investigate the influence of the hyperparameter $\lambda$ on the performance of our proposed method. We varied the value of $\lambda$ among 0.3, 0.5, 0.7, and 0.9, and the corresponding results are presented in Fig. 8(c). The results indicate that larger values of $\lambda$ lead to more effective calibration of GNNs. In our method, $\lambda$ controls the update of the state, which is composed of edge features. A smaller $\lambda$ results in less expressive features, thereby limiting the ability to accurately evaluate the influence of edges on the target in-distribution (ID) nodes.

Finally, we evaluate the sensitivity of our method to the hyperparameters associated with reinforcement learning, specifically focusing on the discount factor $\gamma$ and initial noise $\epsilon_0$. We vary the discount factor among 0.3, 0.5, 0.7, and 0.9, and the initial noise among 0.2, 0.4, 0.6, and 0.8. The corresponding results are presented in Fig. 8(d) and Fig. 8(e), respectively. The results show that calibration performance tends to degrade when using a smaller discount factor or a larger initial noise value.

## 7  Limitations

Although our proposed method effectively calibrates graph neural networks with out-of-distribution (OOD) nodes through an optimized graph structure, it incurs a higher training time compared to baseline methods due to the inherent complexity of reinforcement learning. Nevertheless, by incorporating strategies such as batch processing and sampling of target nodes and edges, the computational cost remains manageable and

does not scale excessively, even on large-scale graphs. Additionally, our current work focuses primarily on calibration for node classification. Extending the approach to other tasks, such as link prediction and graph classification, is an interesting direction for future research.

## 8 Conclusion

In this paper, we focus on calibrating graph neural networks (GNNs) on graphs containing OOD nodes. Noisy graphs exacerbate calibration errors of GNNs, and existing graph calibration methods become less effective. Inspired by graph structure learning, adjusting edge weights presents a plausible solution. However, assigning appropriate edge weights to a noisy graph is a nontrivial task. To address this challenge, we propose the Graph Calibration via Structure Optimization (GCSO) framework to derive an optimal, topology-aware graph structure. Extensive benchmark results demonstrate that our framework effectively calibrates GNNs in the presence of OOD nodes while maintaining comparable accuracy.

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

# A  Appendix

## A.1  Algorithm

---

**Algorithm 1** Algorithm of our GCSO framework

---

**Input:** input graph $\mathcal{G} = (\mathcal{V}, \mathcal{E}, X)$, GNN backbone $f$, labels of the nodes $Y$, candidate nodes set $\mathcal{I}$, critic network $Q(s, a|\theta^Q)$, actor network $\pi(s|\theta^\pi)$, replay buffer $\mathcal{B}$, discount coefficient $\gamma$, hyperparameter $\alpha$, initial noise $\sigma_0$, the total episode $P$, adjacent matrix $\mathbf{A}$.

Initialize the actor network $\pi$, critic network $Q$ and replay buffer $\mathcal{B}$.

**for** 1,2,3..., P **do**

    train the GNN backbone $f$ with adjacent matrix $A$ and obtain the $\text{acc}(B_m)$ and $\text{conf}(B_m)$ on validation nodes. Sample one target node $u$ from the candidate nodes set $\mathcal{I}$.

    obtain the edge set $\mathcal{E}^u = \{e_0^u, e_1^u, \cdots, e_{k-1}^u\}$ for each target node $u$.

    **for** $t$ from 1 to $|\mathcal{E}^u|$ **do**

        calculate the discrepancy score for unsampled edges according to Eq. 4.

        choose the edge $e_t^u$ with the lowest discrepancy score and obtain the state $s_t$ by Eq. 3.

        calculate the adjusted edge weight $w_e = a_t$ from state $s_t$ by Eq. 5.

        add the noise to the adjusted edge weight for exploration via Eq. 8.

        assign the adjusted edge weight to the original graph $\mathcal{G}$.

        obtain the reward $r$ from the GNN backbone $f$ via Eq. 6.

        form the transition tuple $(s_t, a_t, s_{t+1}, r_t)$ into replay buffer $\mathcal{B}$.

        randomly sample the data from replay buffer $\mathcal{B}$ and train the actor network $\pi$ and critic network $Q$ via Eq. 9 and Eq 10.

    **end for**

    generate the new edge weights and obtain new adjacent matrix $A'$ using Eq. 5. Train the GNN backbone $f$ and save the actor and critic networks based on the evaluation of model $f$.

    update the adjacent matrix $\mathbf{A} = \mathbf{A}'$

**end for**

---

Table 9: Comparison between our method with complete reward signal, reward signal without entropy and reward signal without indicator function on Cora, PubMed, Coauthor-CS and Amazon-Computers in terms of node classification accuracy (Acc%)↑ and expected calibration error (ECE%)↓. The experiments are repeated 10 times and the average results and standard deviation are reported. Note that the primary focus of our study is the ECE performance of the methods.

| Methods | Cora | | PubMed | | Coauthor-CS | | Amazon-Computers | |
|---|---|---|---|---|---|---|---|---|
| | Acc | ECE | Acc | ECE | Acc | ECE | Acc | ECE |
| w/o entropy | $84.88 \pm 0.24$ | $10.01 \pm 0.60$ | $\mathbf{92.16} \pm 0.61$ | $1.55 \pm 0.28$ | $\mathbf{92.80} \pm 0.50$ | $2.92 \pm 0.46$ | $90.91 \pm 1.29$ | $3.70 \pm 0.54$ |
| w/o indicator | $84.25 \pm 0.27$ | $10.08 \pm 0.69$ | $92.02 \pm 0.16$ | $1.57 \pm 0.23$ | $92.22 \pm 0.35$ | $3.11 \pm 0.36$ | $90.48 \pm 0.75$ | $3.61 \pm 0.36$ |
| Complete | $\mathbf{84.95} \pm 0.18$ | $\mathbf{9.90} \pm 0.61$ | $\mathbf{92.16} \pm 0.16$ | $\mathbf{1.49} \pm 0.20$ | $92.70 \pm 0.47$ | $\mathbf{2.79} \pm 0.33$ | $\mathbf{91.20} \pm 0.48$ | $\mathbf{3.56} \pm 0.49$ |

## A.2  Experiment

**GCN** (Kipf & Welling, 2016): The learning rate is 1e-2, weight decay is 5e-4. The hidden dimension is 64 with two layers. We choose Adam(Kingma & Ba, 2014) optimizer to train the model.

**CaGCN** (Wang et al., 2021) calibrates the confidence of the GNN by the post-hoc method to ensure the reliability of the prediction, with an estimation of different types of uncertainty. In the experiment we choose GCN (Kipf & Welling, 2016) as the base model. The hidden dimension is 64. The initial learning rate is 1e-2. The number of heads is 8.

**GATS** (Hsu et al., 2022): GATS proposed a new temperature scaling technique to calibrate the graph neural networks. We choose GCN (Kipf & Welling, 2016) as the base model. The hidden dimension is 64. In the experiment. The weight decay is 5e-3.

Table 10: The performance of GKDE-GCN on node classification and OOD node detection with old and new edge weights. The bold represents the best results.

| Dataset | Edge weight | Acc(%) | ECE(%) | OOD AUROC(%) | OOD AUPR(%) |
|---|---|---|---|---|---|
| PubMed | original | **85.61** | 10.73 | 85.21 | **73.58** |
| | Modified | 85.28 | **9.70** | **85.39** | 72.46 |
| Citeseer | original | 65.43 | 4.56 | 80.75 | 81.72 |
| | Modified | **67.93** | **3.56** | **83.41** | **83.57** |
| Amazon Photo | original | 89.83 | 3.05 | 69.05 | 61.35 |
| | Modified | **91.42** | **1.80** | **69.90** | **62.16** |

**GCL** (Wang et al., 2022): GCL loss function is proposed to calibrate the graph neural network in an end-to-end manner. The coefficient $\gamma$ is set to 0.020. The hidden dimension is 64. The rest setting is the same as GCN (Kipf & Welling, 2016).

**OODGAT** (Song & Wang, 2022): OODGAT aims to perform the node classification and OOD detection simultaneously when the graph is mixed with OOD nodes. We adopt the same experimental setting as the original work. Note that the ID/OOD split in our experiment is different from that of OODGAT.

**HyperU-GCN** (Yang et al., 2022) focuses on automated graph learning which can obtain the optimal hyperparameters through joint optimization on model weights and hyperparameters. We adopt the same experimental setting as the original work.

**GERDQ** (Shi et al., 2023) investigates the calibration of graph neural networks when graph is mixed with OOD nodes. GERDQ aims to mitigate the calibration issue by adjusting the edge weight via deep Q-learning (Mnih et al., 2013).

**DCGC** (Yang et al., 2024a) is a data-centric method which aims to calibrate the graph neural networks by assigning larger weights to the decisive and homophilic edges.

In our method, we adopt GCN (Kipf & Welling, 2016) and CaGCN (Wang et al., 2021) and GATS (Hsu et al., 2022) as our GNN backbone. The hyper-parameters of GCN are the same as the corresponding baselines. The learning rate is 1e-2 and weight decay is 5e-4. The hidden dimension is 64. The Actor and Critic in our framework are implemented as a three-layered MLP with the dimension of hidden layers 256 and 16, respectively. Adam (Kingma & Ba, 2014) is adopted for training optimization with the learning rate of 1e-3 for Actor and Critic, The weight decay is 1e-2. The $\alpha$ is set to 0.95, and the discount coefficient $\gamma$ is 0.90. $\sigma_0$ is set to 0.2. The size of the replay buffer is 1e4 and the total number of episodes $P$ is 30. The training epoch is 600 for all datasets. We choose 10 target nodes in the training and then select 16 edges for Cora and Citeseer (Yang et al., 2016), 128 edges for Arxiv (Hu et al., 2020) and 64 edges for the rest. We adopt 10 bins for evaluation of expected calibration error. All the experiments are running on the NVIDIA A5000. We test our method and baselines 10 times with different seeds and the average results are reported.

## A.3 Case Study

We conduct a case study to investigate if the adjusted edge weights can improve the graph learning performance of other methods. GKDE-GCN (Zhao et al., 2020) is a representative method for detecting out-of-distribution (OOD) nodes by uncertainty estimation. We evaluate the performance of GKDE-GCN (Zhao et al., 2020) on node classification and OOD node detection with the adjusted edge weights learned by our framework. The metrics for the OOD node detection are AUROC and AUPR. We run the experiments 10 times on Cora, Citeseer, and PubMed (Yang et al., 2016), and report the average results in Table 10. The results show that our adjusted edge weights can help improve the node classification and OOD detection performance of the base model GKDE-GCN (Zhao et al., 2020).

## A.4 Visualization

To better visualize the ECE, the reliability diagrams of different methods on different datasets are illustrated from Fig. 9 to Fig. 12.

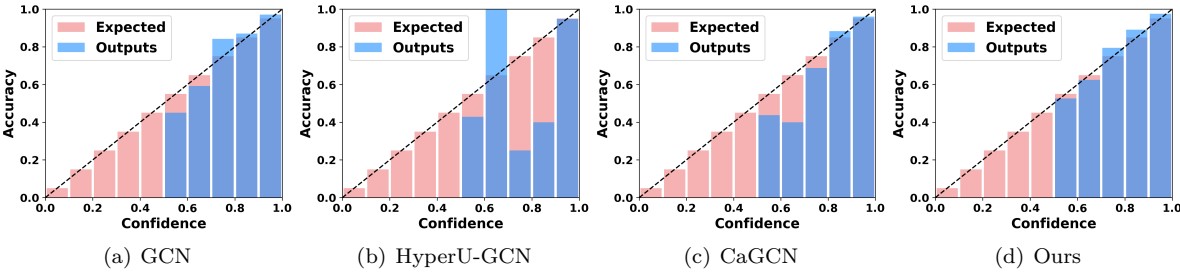

Figure 9: Reliability diagrams of different methods on PubMed with OOD nodes. Well-calibrated results would have closer alignment with the expected results along the diagonal line. The results suggest that the calibration issue is different and complicated on different datasets.

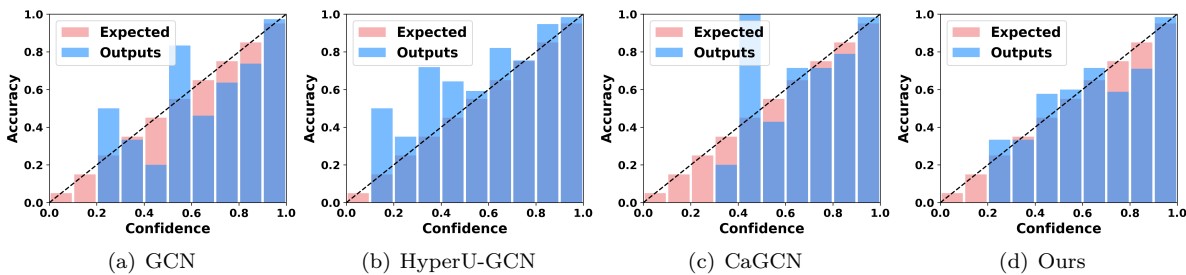

Figure 10: Reliability diagrams of different methods on Coauthor-CS with OOD nodes. Well-calibrated results would have closer alignment with the expected results along the diagonal line. The results suggest that the calibration issue is different and complicated on different datasets.

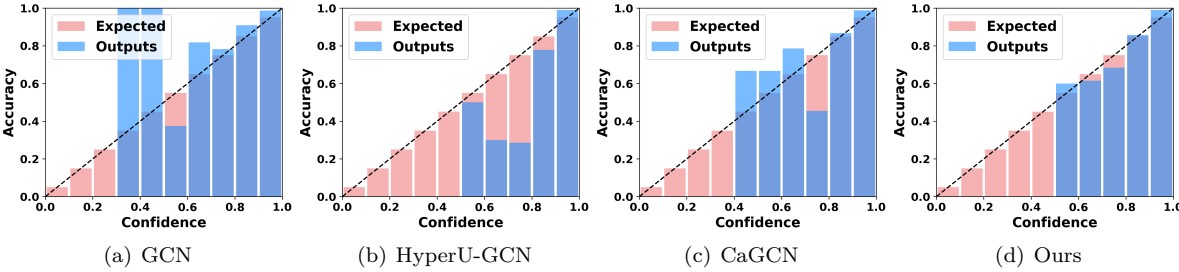

Figure 11: Reliability diagrams of different methods on Coauthor-Physics with OOD nodes. Well-calibrated results would have closer alignment with the expected results along the diagonal line. The results suggest that the calibration issue is different and complicated on different datasets.

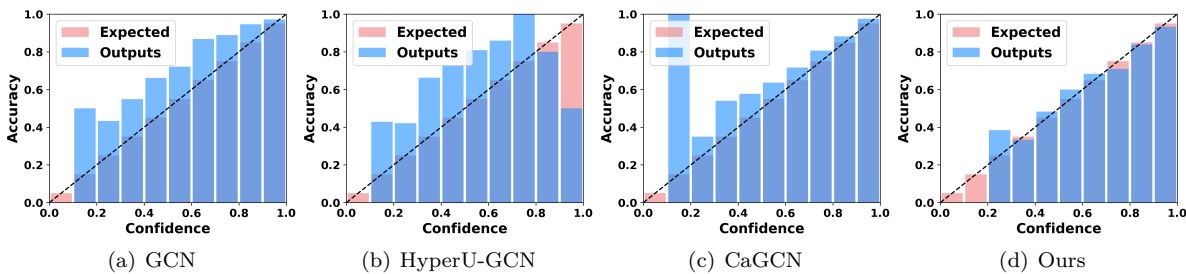

Figure 12: Reliability diagrams of different methods on Arxiv with OOD nodes. Well-calibrated results would have closer alignment with the expected results along the diagonal line. The results suggest that the calibration issue is different and complicated on different datasets.

