# OpenReview forum: "Optimized Graph Structures for Calibrating Graph Neural Networks with Out-of-Distribution Nodes"
_TMLR — Rejected by TMLR_

### Review · Reviewer_Z45L · 2025-04-26

**Summary Of Contributions:**

The paper presents a new method for improving calibration of GNNs in the presence of OOD nodes. The edge weights in the graph are modified during training to assign lower weights to edges connecting OOD nodes with ID nodes. A RL-based framework is introduced for identifying OOD nodes, and finding the optimal edge weights.

**Audience:**

Yes

**Claims And Evidence:**

Yes

**Requested Changes:**

Critical for acceptance:
+ Motivate the use of the RL-based solution for identifying OOD nodes. In fact, the problem of identifying OOD inputs has been extensively studied not just in GNNs, but for inputs of other modalities also. A discussion on these methods, and why they are not effective on graph inputs, will help understand the need for a new OOD detection method.
+ Add results on how effective the proposed method is in identifying OOD nodes. (1) What fraction of OOD nodes are correctly identified in all the benchmarks? (2) What happens if an ID node is incorrectly flagged as a OOD node? (3) How does the proposed method compare to prior works on OOD detection accuracy?
+ Discuss the impact of number of labelled nodes on the calibration gains obtained from the proposed method.
+ Add a discussion on the limitations of the proposed method. In particular,  (1) can this method be used in semi- or self-supervised learning scenarios?, (2) can it be for tasks other than node classification (for example, edge classification or graph classification)?
+ While the main ideas are clearly explained, there are several typos and grammatical errors in the paper. A thorough proof-reading is required before the paper can be accepted.

Would strengthen the work:
+ Add results on semi- and self- supervised learning tasks, and on tasks other than node classification

**Strengths And Weaknesses:**

Strengths:
+ The paper addresses an important problem, since OOD nodes are more problematic for GNNs compared to other neural network architectures. In fact, since predictions in GNNs are not truly independent from each other due to message passing, identifying OOD nodes can improve other metrics also beyond just calibration.
+ The empirical study using a toy example to demonstrate that re-weighting edges can help with calibration is very useful to validate the author's hypothesis, especially since it is presented early in the paper.
+ Code is released to help with reproducibility.

Weaknesses:
+ I could not understand why the RL-based approach is the best choice for this scenario. The authors jump directly into the details of the method without explaining why this approach is the best one for identifying OOD nodes.
+ More ablation studies are required to understand the effectiveness of the proposed RL-based method for OOD detection, and how it compares to previously proposed OOD detection methods.
+ Writing needs to be improved.

---

> ### Author Response · Authors · 2025-06-28
> **Response 1**
>
> **Q1. (Weakness)  I could not understand why the RL-based approach is the best choice for this scenario. The authors jump directly into the details of the method without explaining why this approach is the best one for identifying OOD nodes.
> (Request changes) Motivate the use of the RL-based solution for identifying OOD nodes. In fact, the problem of identifying OOD inputs has been extensively studied not just in GNNs, but for inputs of other modalities also. A discussion on these methods, and why they are not effective on graph inputs, will help understand the need for a new OOD detection method.**
>
> **Response:** Thank you for your valuable suggestion. Identifying out-of-distribution (OOD) inputs is a common challenge across various data modalities, including images and graphs. However, the approaches for OOD detection differ significantly between these domains. In image-based OOD detection, most existing methods rely on confidence scores derived from model outputs to determine whether an input is out-of-distribution. This strategy is effective because image data is typically assumed to be independent and identically distributed (IID). In contrast, graphs are structured data where each node is strongly correlated with its neighbors. As a result, methods designed for image OOD detection often fail when applied to graphs, as they ignore the rich structural dependencies encoded in the graph topology. To better capture the influence of graph structure in identifying OOD nodes, we propose an RL-based approach that optimizes the graph structure by evaluating the impact of individual edges on target in-distribution (ID) nodes. The refined graph structure generated by our method can be seamlessly integrated with existing graph-based OOD detection techniques to further enhance their performance. The relevant discussions have been added to our manuscript.
>
> **Q2. (Weakness) More ablation studies are required to understand the effectiveness of the proposed RL-based method for OOD detection, and how it compares to previously proposed OOD detection methods.
> (Request changes) Add results on how effective the proposed method is in identifying OOD nodes. (1) What fraction of OOD nodes are correctly identified in all the benchmarks? (2) What happens if an ID node is incorrectly flagged as a OOD node? (3) How does the proposed method compare to prior works on OOD detection accuracy?**
>
> **Response:**  Thank you for your comments. We have conducted additional ablation studies to further investigate the effectiveness of our proposed method for OOD detection and compare it against baseline approaches. In our experiments, we use the confidence score to identify OOD nodes, with a threshold set to 0.5. The results on Citeseer, PubMed, and Amazon-Photos are presented in Table 1.
>
> **Table 1**: Comparison of fraction of OOD identified by the baseline and our method on Citeseer, PubMed and Amazon-Photos
> | Method | Citeseer | PubMed | Photos |
> |---|---|---|---|
> | GKDE | 78.46% | 74.24% | 63.18% |
> | Ours | **80.52%** | **78.36%** | **65.76%** |
>
> The results presented in Table 1 demonstrate that our method outperforms the baselines in identifying OOD nodes. In our approach, edges connected to in-distribution (ID) nodes are assigned different weights compared to those connected to OOD nodes. Consequently, if ID nodes are incorrectly classified as OOD, it may lead to inappropriate weight assignments on the associated edges, thereby negatively impacting the calibration of the model on these nodes. The relevant discussions have been added to the manuscript.

---

> ### Author Response · Authors · 2025-06-28
> **Response 2**
>
> **Q3. (Request changes) Discuss the impact of the number of labelled nodes on the calibration gains obtained from the proposed method.**
>
> **Response:** Thank you for your valuable suggestion. We conducted an additional experiment to investigate the influence of the number of labeled nodes on the model’s calibration performance. Specifically, we selected 10, 20, 40, and 80 labeled nodes for training, and conducted evaluations on the PubMed and Coauthor-CS datasets. The results are presented in Table 2.
>
> **Table 2**: Performance of our method on node classification and calibration with varying numbers of labelled nodes on PubMed and Coauthor-CS
> | Dataset | #node (10) | | #node (20) | | #node (40) | | #node (80) | |
> |---|---|---|---|---|---|---|---|---|
> | | Acc | ECE | Acc | ECE | Acc | ECE | Acc | ECE |
> | PubMed | 92.16 ± 0.16 | 1.49 ± 0.20 | 92.34 ± 0.34 | 1.42 ± 0.39 | 92.14 ± 0.18 | 1.40 ± 0.23 | 92.33 ± 0.16 | 1.36 ± 0.28 |
> | CS | 91.71 ± 0.33 | 2.56 ± 0.17 | 91.96 ± 0.25 | 2.47 ± 0.13 | 91.87 ± 0.30 | 2.43 ± 0.10 | 91.90 ± 0.31 | 2.45 ± 0.12 |
>
> The results suggest that increasing the number of labeled nodes leads to an improvement in calibration error. A larger number of labeled nodes allows for more precise evaluation of the influence of individual edges on the labeled nodes. With more labeled samples, the model can better optimize the graph structure, resulting in improved calibration performance.
>
> **Q4. (Request changes) Add a discussion on the limitations of the proposed method. In particular, (1) can this method be used in semi- or self-supervised learning scenarios?, (2) can it be for tasks other than node classification (for example, edge classification or graph classification)?
> Would strengthen the work:
> Add results on semi- and self- supervised learning tasks, and on tasks other than node classification**
>
> **Response:** Thank you for your comments. Our work builds upon prior research that focuses on calibration in graph-based models. Specifically, both the previous work and our approach adopt a semi-supervised learning setting, where the model is trained with a limited number of labeled nodes and the primary objective is to improve calibration in node classification. Since our method is specifically designed for this setting, it may not be directly applicable to self-supervised learning or to other tasks such as edge classification and graph classification. Extending our approach to these settings is an interesting direction for future work. The relevant discussions have been added to the manuscript.
>
> **Q5. (Weakness) Writing needs to be improved.
> (Request changes) While the main ideas are clearly explained, there are several typos and grammatical errors in the paper. A thorough proof-reading is required before the paper can be accepted.**
>
> **Response:** Thank you for your suggestion. We have carefully proofread the manuscript and revised the typographical and grammatical errors accordingly. All revisions are highlighted in the manuscript.

---

### Review · Reviewer_QNtW · 2025-05-27

**Summary Of Contributions:**

This paper proposes GCSO, a calibration framework for GNNs in scenarios where OOD nodes are mixed within the input graph. To mitigate negative impacts from OOD nodes, this method dynamically adjusts edge weights corresponding to the distribution of OOD nodes, aiming to improve calibration and inference accuracy.

**Audience:**

Yes

**Claims And Evidence:**

No

**Requested Changes:**

- Improve the overall presentation: address grammatical errors, refine sentence structures, and enhance logical flow.
- Replace strong claims with a weaker expression or provide theoretical justification to support these claims.
- Expand the experimental validation to include more varied and generalizable dataset configurations, explicitly addressing different OOD node distributions.
- (Question) This paper shows similar ECE values for GCN and GERDQ, whereas the original GERDQ paper demonstrated a significant reduction in ECE compared to GCN. Please clarify the reason for this discrepancy.

Also, considering the weaknesses section presented above, please change the manuscript accordingly.

**Strengths And Weaknesses:**

**Strengths:**
- It is motivated by empirical observations and practical relevance.
- Visual illustrations (figures) aid understanding.

**Weaknesses:**
- Overall low presentation quality: numerous grammatical and structural errors impede readability and clarity. Some sections remain unclear or redundant (e.g., Section 4).
- Although the paper states, "First, our method adopts an indicator function in the reward signal to ensure the adjusted graph structure would not compromise the accuracy on the ID nodes." the theoretical support for this claim is unclear and lacks discussion.
- Empirical validation is limited and not particularly compelling. The performance of the proposed method alone does not appear superior to that of conventional methods.
- Given the current experimental setup, specifically the choice of OOD nodes (e.g., selecting classes 5 and 6 as OOD nodes in the Cora dataset), it is difficult to conclude the generalizability and robustness of the results.
- Figure 6 suggests the limited effectiveness of the proposed method compared to existing methods such as CaGCN.

---

> ### Author Response · Authors · 2025-06-28
> **Response 1**
>
> **Q1.(Weakness) Overall low presentation quality: numerous grammatical and structural errors impede readability and clarity. Some sections remain unclear or redundant (e.g., Section 4).
> (Request changes) Improve the overall presentation: address grammatical errors, refine sentence structures, and enhance logical flow.**
>
> **Response:** Thank you for your suggestion. We have carefully proofread the manuscript and revised the grammatical errors and refine the structure to improve the presentation. All revisions are highlighted in the manuscript.
>
> **Q2.(Weakness)  Although the paper states, "First, our method adopts an indicator function in the reward signal to ensure the adjusted graph structure would not compromise the accuracy on the ID nodes." the theoretical support for this claim is unclear and lacks discussion.
> (Request changes) Replace strong claims with a weaker expression or provide theoretical justification to support these claims.**
>
> **Response:** Thanks for your valuable suggestion. In response to your suggestion, we have revised the original claim to adopt a more cautious expression. The statement now reads:
> "Since the modified graph structure may affect the quality of the learned representations, and thereby influence the performance on downstream tasks such as node classification, we incorporate an indicator function in the reward signal. This component empirically encourages the preservation of classification accuracy on the ID nodes during graph structure optimization."
> We have updated the manuscript accordingly to reflect this change.
>
> **Q3. (Weakness)  Empirical validation is limited and not particularly compelling. The performance of the proposed method alone does not appear superior to that of conventional methods.**
>
> **Response:** Thank you for your feedback. Calibrating graph neural networks (GNNs) is particularly challenging in the presence of out-of-distribution (OOD) nodes. In some cases, conventional calibration methods may even worsen the calibration error. For example, on the PubMed dataset, the expected calibration error (ECE) of a standard GCN is around 1.59%, while methods such as CaGCN yield a higher ECE of approximately 3.09% on average. This suggests that existing methods may fail to effectively calibrate GNNs when OOD nodes are present. In contrast, our proposed method is able to mitigate the negative influence of OOD nodes and achieve a lower calibration error of 1.49% on the same dataset. A similar phenomenon can also be observed on other benchmarks, such as Coauthor-CS. Furthermore, we highlight that our method can be seamlessly integrated with existing post-hoc calibration methods to further reduce the calibration error of GNNs.

---

> > ### Author Response · Authors · 2025-06-28
> > **Response 3**
> >
> > **Q5. (Weakness) Figure 6 suggests the limited effectiveness of the proposed method compared to existing methods such as CaGCN.**
> >
> > **Response:** Thank you for your comments. Figure 6 presents the reliability diagrams to visually demonstrate the discrepancy between model confidence and the actual likelihood of correctness. Ideally, for a well-calibrated model, the blue bars (accuracy) should align closely with the red bars (confidence). The difference between these bars reflects the calibration error. As shown in Figure 6, our proposed method exhibits a smaller calibration gap compared to CaGCN, indicating better calibration performance. Additional visualizations are provided in Figures 8–11 for further comparison.
> >
> > **Q6. (Request changes) This paper shows similar ECE values for GCN and GERDQ, whereas the original GERDQ paper demonstrated a significant reduction in ECE compared to GCN. Please clarify the reason for this discrepancy.**
> >
> > **Response:** Thank you for your comments. The discrepancy in the results may be attributed to two main factors. First, the OOD node partitions differ between our work and the original GERDQ paper. In GERDQ, nodes from half of the classes are treated as OOD. For example, in Amazon-Computers with 10 classes, nodes from classes 5 to 9 are considered OOD. However, this setting may not accurately reflect real-world scenarios, where OOD nodes typically constitute a small fraction of the overall node population. Therefore, in our paper, we adopt a more realistic configuration by selecting nodes from only a few specific classes as OOD. For instance, in Amazon-Computers, we designate nodes from classes 7 to 9 as OOD nodes. Second, we follow the practice in prior work [1] by using different random seeds to generate training, validation, and testing splits, which ensures more robust evaluation. In contrast, the GERDQ paper uses fixed splits across experiments. This difference in data partitioning may also contribute to the observed variation in results.
> >
> >
> > [1] Hsu, Hans Hao-Hsun, et al. "What makes graph neural networks miscalibrated?." Advances in Neural Information Processing Systems 35 (2022): 13775-13786.

---

> ### Author Response · Authors · 2025-06-28
> **Response 2**
>
> **Q4.  (Request changes) Expand the experimental validation to include more varied and generalizable dataset configurations, explicitly addressing different OOD node distributions.**
>
> **Response:** Thank you for your valuable suggestion. To further assess the generalizability and robustness of our proposed method, we expanded our experiments by evaluating it on datasets with different OOD node distributions. The OOD configurations are detailed in Table 1, and the results are reported in Tables 2–4.
>
> **Table 1**: The OOD configurations of Cora, PubMed and Computers
> | Dataset | Config 1 | | Config 2 | | Config 3 | |
> |---|---|---|---|---|---|---|
> | | ID classes | OOD classes | ID classes | OOD classes | ID classes | OOD classes |
> | Cora | [0-3,6] | [1,2] | [0-2,5,6] | [3,4] | [0-4] | [5,6] |
> | PubMed | [1,2] | [0] | [0,2] | [1] | [0,1] | [2] |
> | Computers | [0-4,9] | [1,2,3] | [0-3,7-9] | [4,5,6] | [0-6] | [7,8,9] |
>
> **Table 2**: Comparison between our proposed method and baselines in terms of node classification accuracy (Acc%) and expected calibration error (ECE%) on Cora with different OOD configurations.
> | Methods | Cora (Config 1) | | Cora (Config 2) | | Cora (Config 3) | |
> |---|---|---|---|---|---|---|
> | | Acc | ECE | Acc | ECE | Acc | ECE |
> | GCN | 81.91±0.57 | 9.88±0.71 | 84.85±0.22 | 9.42±0.75 | 84.18±0.28 | 9.90±0.61 |
> | CaGCN | 81.88±0.50 | 4.10±0.44 | 84.82±0.26 | 3.39±0.57 | 84.14±0.35 | 3.85±1.05 |
> | GATS | **82.61±0.71** | 4.11±0.62 | 85.42±0.54 | 3.03±1.13 | 83.49±0.31 | 2.81±0.82 |
> | DCGC | 80.86±0.18 | 8.62±0.70 | 83.16±0.44 | 8.21±0.48 | 82.46±0.26 | 8.57±0.82 |
> | GERDQ | 82.02±0.59 | 9.46±0.60 | 84.53±0.46 | 9.25±0.35 | 83.67±0.48 | 9.54±0.50 |
> | GCSO | 81.86±0.56 | 9.27±0.43 | 84.77±0.23 | 9.11±0.17 | **84.95±0.18** | 9.22±0.49 |
> | GCSO+CaGCN | 82.14±0.42 | 3.75±0.38 | 84.80±0.28 | 2.83±0.38 | 84.28±0.27 | **2.55±0.45** |
> | GCSO+GATS | 81.89±0.54 | **2.49±0.33** | **85.46±0.78** | **2.78±0.49** | 84.20±0.31 | 2.63±0.46 |
>
> **Table 3**: Comparison between our proposed method and baselines in terms of node classification
>  accuracy (Acc%) and expected calibration error (ECE%) on PubMed with different OOD configurations.
> | Methods | PubMed (Config 1) | | PubMed (Config 2) | | PubMed (Config 3) | |
> |---|---|---|---|---|---|---|
> | | Acc | ECE | Acc | ECE | Acc | ECE |
> | GCN | 83.65±0.27 | 3.95±0.76 | 90.32±0.30 | 1.77±0.74 | 92.11±0.17 | 1.59±0.56 |
> | CaGCN | 83.67±0.28 | 5.07±0.50 | 90.32±0.31 | 1.64±0.44 | 92.11±0.17 | 3.09±0.20 |
> | GATS | 83.47±0.32 | 3.67±0.58 | 89.81±0.30 | 1.47±0.60 | 92.56±0.24 | 2.27±0.39 |
> | DCGC | 83.39±0.41 | 3.68±0.70 | 89.88±0.29 | 2.41±0.33 | 92.66±0.16 | 2.43±0.44 |
> | GERDQ | 83.65±0.27 | 3.69±0.57 | 90.32±0.30 | 1.62±0.61 | 92.14±0.20 | 1.60±0.59 |
> | GCSO | **83.73±0.28** | 3.68±0.51 | **90.39±0.32** | 1.36±0.43 | 92.16±0.16 | **1.49±0.20** |
> | GCSO+CaGCN | 83.70±0.26 | 4.86±0.55 | **90.39±0.30** | 1.60±0.21 | 92.24±0.26 | 2.80±0.19 |
> | GCSO+GATS | 83.52±0.45 | **3.42±0.39** | 89.69±0.40 | **1.30±0.19** | **92.69±0.27** | 2.13±0.34 |
>
> **Table 4**: Comparison between our proposed method and baselines in terms of node classification
>  accuracy (Acc%) and expected calibration error (ECE%) on Computers with different OOD configurations.
> | Methods | Computers (Config 1) | | Computers (Config 2) | | Computers (Config 3) | |
> |---|---|---|---|---|---|---|
> | | Acc | ECE | Acc | ECE | Acc | ECE |
> | GCN | 87.24 ± 0.58 | 2.74 ± 0.40 | **93.79 ± 0.39** | 2.61 ± 0.23 | 90.81 ± 0.53 | 3.02 ± 0.68 |
> | CaGCN | 88.14 ± 0.24 | 4.41 ± 0.54 | 92.25 ± 0.42 | 3.16 ± 0.35 | 88.67 ± 0.38 | 2.82 ± 0.17 |
> | GATS | 87.60 ± 0.77 | 3.21 ± 0.53 | 92.25 ± 0.49 | 2.75 ± 0.41 | 88.07 ± 0.43 | 3.59 ± 0.68 |
> | DCGC | 87.90 ± 0.36 | 3.32 ± 0.71 | 93.64 ± 0.55 | 2.66 ± 0.41 | 90.84 ± 0.66 | 2.49 ± 0.43 |
> | GERDQ | 87.08 ± 0.54 | 2.60 ± 0.23 | 93.41 ± 0.65 | 2.76 ± 0.34 | 90.52 ± 0.42 | 2.52 ± 0.49 |
> | GCSO | 87.17 ± 0.41 | **2.56 ± 0.35** | 93.16 ± 0.52 | 2.58 ± 0.16 | **90.86 ± 0.45** | **2.44 ± 0.28** |
> | GCSO+CaGCN | **88.30 ± 0.56** | 4.03 ± 0.45 | 92.56 ± 0.48 | 2.84 ± 0.19 | 88.50 ± 0.30 | 2.64 ± 0.29 |
> | GCSO+GATS | 87.65 ± 0.75 | 3.07 ± 0.30 | 92.17 ± 0.35 | **2.40 ± 0.21** | 88.49 ± 0.71 | 3.33 ± 0.34 |
>
> The results suggest that conventional methods such as CaGCN struggle to calibrate GNNs under varying OOD configurations. For example, on the Amazon-Computers dataset, CaGCN yields higher calibration errors in both Configuration 1 (4.41%) and Configuration 2 (3.16%) compared to the baseline GCN. This failure may stem from the method's disregard for graph topology, particularly the distribution of OOD nodes, when adjusting the output logits. In contrast, our method consistently improves calibration across different OOD settings. The improvements are more pronounced on large graphs (e.g., PubMed and Computers) than on smaller graphs (e.g., Cora). These results across three datasets empirically demonstrate the generalizability and robustness of our approach under diverse OOD distributions.

---

### Review · Reviewer_Ge3f · 2025-06-14

**Summary Of Contributions:**

The core contribution of this paper is the development of the GCSO framework, a data-centric approach to calibrate GNNs in the presence of OOD nodes. The key innovations are:

1. RL-based Edge Re-weighting: The process of adjusting edge weights is formulated as a Markov Decision Process (MDP). An actor-critic reinforcement learning method, specifically Deep Deterministic Policy Gradient (DDPG), is used to learn a policy that outputs fine-grained, topology-aware edge weights.
2. Iterative Edge Sampling: Instead of random sampling, the framework iteratively selects edges to re-weight based on a "discrepancy score," which measures the KL divergence between the logit distributions of connected nodes. This allows the model to assess the impact of edges connecting dissimilar nodes strategically.
3. Specialized Reward Signal: A novel reward function is designed to optimize for calibration directly.
In addition, the optimized graph structure generated by GCSO can be seamlessly integrated with existing post-hoc calibration methods (e.g., Temperature Scaling-based techniques like CaGCN and GATS) to achieve further performance gains.

**Audience:**

Yes

**Broader Impact Concerns:**

No.

**Claims And Evidence:**

Yes

**Requested Changes:**

Please see the weakness part above. While I do not expect a new set of experiments, I would appreciate it if you could explain every potential weakness part as clearly as possible, thanks.

**Strengths And Weaknesses:**

Strengths
1.  The paper targets a highly relevant and under-explored topic: GNN calibration specifically in the OOD context. The initial empirical study effectively highlights that this is a non-trivial problem distinct from standard calibration.
2. The use of an actor-critic RL framework is a powerful and fitting choice for learning continuous edge weights. The formulation as an MDP is clear and logical. The design of the reward signal (Eq. 6) is the most impressive aspect of this work. By explicitly incorporating a term to correct for over/under-confidence, the framework directly targets the calibration problem, rather than hoping that accuracy improvements will indirectly lead to better calibration.

While the paper is innovative and new in several aspects, I still have some comments on the potential weakness of this paper:
1. The primary weakness is the computational overhead. The authors acknowledge that the training time is greater than that of baselines like DCGC. While they propose batching and limiting the number of sampled nodes/edges as a solution, the RL-based training loop is inherently more complex and time-consuming than standard GNN training. This could be a significant barrier for applying GCSO to extremely large-scale industrial graphs.
2. The policy function is trained for a specific graph. As the authors note, it needs to be retrained for different graphs due to varying topological structures. This limits the "plug-and-play" nature of the solution and requires a full training cycle for each new graph dataset.
3. The framework introduces several new hyperparameters, including the state-update balancing factor λ , RL-specific parameters (e.g., noise, discount factor), and the number of target nodes and edges for iteration. The paper does not include a sensitivity analysis, making it difficult to gauge how robust the method is to these choices and how much tuning is required to achieve the reported results.
4. The reward signal, which is critical for learning, depends on labeled validation nodes to determine over/under-confidence bins and calculate the indicator function. The performance might degrade in semi-supervised settings with very few labeled nodes, which are common in graph-based tasks

---

> ### Author Response · Authors · 2025-06-27
> **Response 1**
>
> **Q1.(Weakness) The primary weakness is the computational overhead. The authors acknowledge that the training time is greater than that of baselines like DCGC. While they propose batching and limiting the number of sampled nodes/edges as a solution, the RL-based training loop is inherently more complex and time-consuming than standard GNN training. This could be a significant barrier for applying GCSO to extremely large-scale industrial graphs.**
>
> **Response:**  Thank you for your comment. To alleviate computational overhead, we adopt two techniques: batch processing and sampling of nodes/edges. Our empirical results suggest that even with a very limited number of sampled nodes and edges, the calibration issue can still be effectively mitigated. For instance, on the Amazon-Computers dataset, GCN achieves an average ECE of 3.02%, while our method achieves approximately 2.44% when trained with only 20 labeled nodes. Thanks to these strategies, the computational cost does not escalate, even when handling extremely large-scale graphs, and remains within a reasonable range.
>
> **Q2: (Weakness) The policy function is trained for a specific graph. As the authors note, it needs to be retrained for different graphs due to varying topological structures. This limits the "plug-and-play" nature of the solution and requires a full training cycle for each new graph dataset.**
>
> **Response:** Thank you for your comment. Calibrating graph neural networks (GNNs) is a challenging task, particularly in the presence of out-of-distribution (OOD) nodes. In such scenarios, both the topological structure and the distribution of OOD nodes play crucial roles in achieving effective calibration. Since these factors can vary significantly across different graphs, our method trains a policy function to optimize the graph structure for each specific graph. Once the optimized graph structure is obtained, it can be seamlessly integrated with existing post-hoc calibration methods (e.g., CaGCN) to further reduce calibration error. This demonstrates the plug-and-play nature of our approach.

---

> ### Author Response · Authors · 2025-06-28
> **Response_2**
>
> **Q3: (Weakness) The framework introduces several new hyperparameters, including the state-update balancing factor λ , RL-specific parameters (e.g., noise, discount factor), and the number of target nodes and edges for iteration. The paper does not include a sensitivity analysis, making it difficult to gauge how robust the method is to these choices and how much tuning is required to achieve the reported results.**
>
> **Response:**
> **Table 1**: Performance of our method on node classification and calibration with varying numbers of labelled nodes on PubMed and Coauthor-CS.
> | Dataset | #node (10) | | #node (20) | | #node (40) | | #node (80) | |
> |---|---|---|---|---|---|---|---|---|
> | | Acc | ECE | Acc | ECE | Acc | ECE | Acc | ECE |
> | PubMed | 92.16 ± 0.16 | 1.49 ± 0.20 | 92.34 ± 0.34 | 1.42 ± 0.39 | 92.14 ± 0.18 | 1.40 ± 0.23 | 92.33 ± 0.16 | 1.36 ± 0.28 |
> | CS | 91.71 ± 0.33 | 2.56 ± 0.17 | 91.96 ± 0.25 | 2.47 ± 0.13 | 91.87 ± 0.30 | 2.43 ± 0.10 | 91.90 ± 0.31 | 2.45 ± 0.12 |
>
> **Table 2**: Performance of our method on node classification and calibration with varying numbers of edges on PubMed and Coauthor-CS.
> | Dataset | #edge (10) | | #edge (20) | | #edge (40) | | #edge (60) | |
> |---|---|---|---|---|---|---|---|---|
> | | Acc | ECE | Acc | ECE | Acc | ECE | Acc | ECE |
> | PubMed | 92.23 ± 0.22 | 1.55 ± 0.39 | 92.16 ± 0.16 | 1.49 ± 0.20 | 92.14 ± 0.28 | 1.43 ± 0.29 | 92.39 ± 0.22 | 1.40 ± 0.23 |
> | CS | 91.94 ± 0.31 | 2.59 ± 0.10 | 91.88 ± 0.36 | 2.60 ± 0.16 | 91.96 ± 0.25 | 2.47 ± 0.13 | 91.93 ± 0.29 | 2.41 ± 0.13 |
>
> **Table 3**: Performance of our method on node classification and calibration with varying value of $\lambda$ on PubMed and Coauthor-CS.
> | Dataset | $\lambda$ (0.3) | | $\lambda$ (0.5) | | $\lambda$ (0.7) | | $\lambda$ (0.9) | |
> |---|---|---|---|---|---|---|---|---|
> | | Acc | ECE | Acc | ECE | Acc | ECE | Acc | ECE |
> | PubMed | 92.43 ± 0.22 | 1.71 ± 0.24 | 92.41 ± 0.28 | 1.66 ± 0.31 | 92.45 ± 0.31 | 1.56 ± 0.24 | 92.16 ± 0.16 | 1.49 ± 0.20 |
> | CS | 91.93 ± 0.37 | 2.69 ± 0.13 | 91.88 ± 0.33 | 2.52 ± 0.13 | 91.88 ± 0.34 | 2.53 ± 0.08 | 91.96 ± 0.25 | 2.47 ± 0.13 |
>
> **Table 4**: Performance of our method on node classification and calibration with varying value of discount factor on Coauthor-CS.
> | Dataset | discount (0.3) | |  discount  (0.5) | |  discount  (0.7) | |   discount (0.9) | |
> |---|---|---|---|---|---|---|---|---|
> | | Acc | ECE | Acc | ECE | Acc | ECE | Acc | ECE |
> | CS | 91.82 ± 0.32 | 2.55 ± 0.14 | 91.85 ± 0.33 | 2.53 ± 0.15 | 91.90 ± 0.33 | 2.49 ± 0.10 | 91.96 ± 0.25 | 2.47 ± 0.13 |
>
> **Table 5**: Performance of our method on node classification and calibration with varying value of initial noise on Coauthor-CS.
> | Dataset | noise (0.2) | | noise  (0.4) | |  noise  (0.6) | | noise  (0.8) | |
> |---|---|---|---|---|---|---|---|---|
> | | Acc | ECE | Acc | ECE | Acc | ECE | Acc | ECE |
> | CS | 91.96 ± 0.25 | 2.47 ± 0.13 | 91.92 ± 0.29 | 2.49 ± 0.08 | 91.96 ± 0.31 | 2.52 ± 0.09 | 91.93 ± 0.34 | 2.53 ± 0.16 |
>
> Thank you for your suggestion. To evaluate the impact of the number of sampled target nodes and edges on the performance of our method, we conducted additional experiments on the PubMed and Coauthor-CS datasets. In the first experiment, we fixed the number of sampled edges to 20 for PubMed and 40 for Coauthor-CS, and evaluated the performance using 10, 20, 40, and 80 sampled nodes. In the second experiment, we fixed the number of sampled nodes to 10 for PubMed and 20 for Coauthor-CS, and varied the number of sampled edges among 10, 20, 40, and 60. The corresponding results are presented in Table 1 and Table 2. These results indicate that increasing the number of sampled nodes and edges generally improves calibration performance. However, beyond a certain point, the improvements become marginal while the computational cost increases.
>
> We also conducted an evaluation to investigate the influence of the hyperparameter λ on the performance of our proposed method. We varied the value of λ among 0.3, 0.5, 0.7, and 0.9, and the corresponding results are presented in Table 3. The results indicate that larger values of λ lead to more effective calibration of GNNs. In our method, λ controls the update of the state, which is composed of edge features. A smaller λ results in less expressive features, thereby limiting the ability to accurately evaluate the influence of edges on the target in-distribution (ID) nodes.
>
> Finally, we evaluate the sensitivity of our method to the hyperparameters associated with reinforcement learning, specifically focusing on the discount factor and initial noise. We vary the discount factor among 0.3, 0.5, 0.7, and 0.9, and the initial noise among 0.2, 0.4, 0.6, and 0.8. The corresponding results are presented in Table 4 and Table 5. The results show that calibration performance tends to degrade when using a smaller discount factor or a larger initial noise value. These findings have been incorporated into the revised manuscript.

---

> ### Author Response · Authors · 2025-06-28
> **Response 3**
>
> **Q4: (Weakness)  The reward signal, which is critical for learning, depends on labeled validation nodes to determine over/under-confidence bins and calculate the indicator function. The performance might degrade in semi-supervised settings with very few labeled nodes, which are common in graph-based tasks.**
>
> **Response:** Thank you for your comment. Our method relies on labeled nodes to obtain the reward signal used for optimizing the graph structure, and the number of sampled labeled nodes plays a significant role in this process. In semi-supervised settings, only a limited number of labels are available. However, as shown in Table 1, our method can still effectively mitigate the calibration issue even with a small number of labeled nodes, demonstrating its effectiveness in semi-supervised scenarios.

---

### Decision · Action_Editor_zAa7 · 2025-08-01

**Recommendation:** Reject

**Additional Comments:**

N/A

**Audience:**

Yes

**Audience Explanation:**

Graph learning is an important direction in the research community of AI, and its role is critical even in the context of foundation models as the basic knowledge source. Many researchers pay attention to the related topics, which should be interested in this manuscript's study.

**Claims And Evidence:**

No

**Claims Explanation:**

The submission received comments of three reviewers. In the first-round review, the reviewers provided several constructive suggestions and concerns, and the authors provided substantial improvement. However, after rebuttal, there are still some remaining concerns that make some reviewer unconvinced, yielding "reject" recommendation. I listed some points of concerns as follows,

> The existing and additional results do not demonstrate that the proposed method, GCSO, outperforms the existing approaches. For example, the authors emphasize that GCSO achieves better performance on the PubMed dataset (table 3 in the responce), but the improvements are marginal. Given the large standard deviations, it is likely that with a different random seed, the performance advantage may disappear. Moreover, GCSO alone does not perform particularly well; its slight improvement over baselines seems to rely heavily on being combined with the existing methods. This result suggests that the effectiveness of GCSO is largely dependent on the existing methods.
> Due to these concerns, I do not have a positive impression of this proposed method. I understand that TMLR does not require state-of-the-art performance for acceptance, but I think that a method which consistently fails to show statistically meaningful improvements in any experimental setting should not be accepted.
>Additionally, the extent of the revisions is substantial (approximately three pages of additional content), which gives the impression of the authors merely appending responses. In my opinion, it would be more appropriate for the authors to reconstruct the paper and resubmit it in a more polished and coherent form.

The AE carefully checked three reviewers' recommendation and their response to the rebuttal, and tends to agree with the overall recommendation, and consider that there are some points requiring improvement for this manuscript, yielding "reject".